# Hypocretinergic interactions with the serotonergic system regulate REM sleep and cataplexy

Ali Seifinejad [1], Sha Li[1], Marie-Laure Possovre[1], Anne Vassalli [1,2] & Mehdi Tafti [1,2 ✉]

Loss of muscle tone triggered by emotions is called cataplexy and is the pathognomonic symptom of narcolepsy, which is caused by hypocretin deficiency. Cataplexy is classically considered to be an abnormal manifestation of REM sleep and is treated by selective serotonin (5HT) reuptake inhibitors. Here we show that deleting the 5HT transporter in hypocretin knockout mice suppressed cataplexy while dramatically increasing REM sleep. Additionally, double knockout mice showed a significant deficit in the buildup of sleep need. Deleting one allele of the 5HT transporter in hypocretin knockout mice strongly increased EEG theta power during REM sleep and theta and gamma powers during wakefulness. Deleting hypocretin receptors in the dorsal raphe neurons of adult mice did not induce cataplexy but consolidated REM sleep. Our results indicate that cataplexy and REM sleep are regulated by different mechanisms and both states and sleep need are regulated by the hypocretinergic input into 5HT neurons.

[1] Department of Biomedical Sciences, Faculty of Biology and Medicine, University of Lausanne, 1005 Lausanne, Switzerland. [2] These authors contributed equally: Anne Vassalli, Mehdi Tafti. ✉email: mehdi.tafti@unil.ch

A constellation of cell nuclei located in the brainstem, hypothalamus, basal forebrain, thalamus and cortex communicate with each other to fine-tune the complex behavior sleep. Neurotransmitters and neuropeptides released from these nuclei are wake- or sleep-promoting, and their interactions lead to electroencephalographically (EEG) measurable readouts, which are wakefulness, non-rapid eye movement sleep (NREMS) and rapid eye movement sleep (REMS). NREMS typically cycles with REMS and wakefulness. This wake-NREM-REMS cycle is dysregulated in the sleep disorder narcolepsy.

Narcolepsy with cataplexy (also called narcolepsy type 1 or NT1) is characterized by excessive daytime sleepiness, cataplexy (sudden loss of muscle tone during wakefulness), hypnogogic hallucinations and sleep paralysis[1]. The cataplexy and sleep paralysis were interpreted as representing the coexistence of the muscle atonia of REMS with preserved consciousness of wakefulness. REMS regulation is also impaired in narcolepsy patients who experience multiple sleep-onset REM periods during daytime naps, which are rare in normal subjects[2].

A key wake-promoting system called orexin or hypocretin (hereafter HCRT) is disabled in patients with narcolepsy who presumably lost their HCRT producing neurons[3]. The most prevalent hypothesis suggests an autoimmune attack against HCRT neurons[4–6]. HCRTs (HCRT-1 and -2) are small neuropeptides exclusively produced by a neuronal population of the lateral hypothalamus (LH)[7,8]. Although confined to the LH, HCRT neurons widely project to the entire central nervous system and signal through two receptors, HCRTR1 and HCRTR2[7]. Wake-promoting nuclei of the brainstem and basal forebrain are major targets of the HCRT system[9]. Animal studies established that the HCRT system is critical in sustaining the normal waking state and its genetic defects lead to narcolepsy phenotypes in mice and dogs[10,11].

Symptoms of narcolepsy with cataplexy are treated with stimulants and antidepressants[1]. Stimulants increase catecholamines (dopamine and noradrenalin) or histamine, while most antidepressants increase serotonin (5-hydroxytryptamine, 5HT). Dorsal raphe (DR) 5HT neurons are active during wakefulness, decrease activity during NREMS, and are mostly silent during REMS[12]. In contrast to noradrenergic cells of the locus coeruleus, 5HT neurons do not completely cease firing during cataplexy[12]. These cells project throughout the brain and functions by releasing 5HT, which binds 5HT receptors (up to 14 different 5HT receptors)[13]. The action of 5HT is terminated by its reuptake from the synaptic cleft into the presynaptic terminals, a process mediated by the serotonin transporter (SERT or 5HTT, encoded by the Slc6a4 gene). 5HTT is the target of major antidepressants called selective serotonin reuptake inhibitors (SSRI), used to treat cataplexy in narcolepsy patients[14].

The DR-5HT system was recently proposed to be intimately linked to sleep promotion and sleep pressure in a bidirectional process dependent on 5HT cells' firing mode[15]. Their higher activity during wakefulness was suggested to be part of a homeostatic process serving to build-up sleep pressure[16]. Although original lesions of the DR nucleus or the suppression of its activity did not result in changes in REMS amount, the 5HT system is still involved in the regulation of REMS since null mutations of either Htr1a or Htr1b genes and pharmacological blockade of these receptors result in increased REMS[17,18]. Furthermore, mice lacking functional 5HTT exhibit an increased amount of REMS[19] indicating that intact 5HT transmission is required to fine-tune REMS.

DR is one of the major targets of HCRT axonal projections. Restoring expression of HCRT receptors 1&2 in the DR of mice deficient in both receptors rescues cataplexy[20]. SSRIs increase the brain serotonin tone by inhibiting serotonin reuptake and this increase is hypothesized to prevent cataplexy attacks[21]. Therefore, we hypothesized that introducing a null mutation of 5HTT gene (Slc6a4)[22] in Hcrt knockout mice[10] might suppress cataplexy and normalize their sleep architecture. To this end, we investigated vigilance states and cataplexy in mice lacking both Slc6a4 and Hcrt genes. We also asked if the removal of HCRT receptors 1&2 from DR serotonergic neurons has an impact on the expression of cataplexy and normal vigilance states distribution.

## Results

To introduce 5HTT deficiency in mice lacking hypocretins, heterozygous mice for Hcrt gene ($Hcrt^{+/ko}$) were mated with heterozygous 5HTT mice ($Slc6a4^{+/KO}$ hereafter $5HTT^{+/ko}$). This cross generates nine genotypes, of which five littermate offspring groups were analyzed: wild type (WT), Hcrt null ($Hcrt^{KO/KO}$), 5HTT null ($5HTT^{KO/KO}$), $5HTT^{+/KO};Hcrt^{KO/KO}$, and Hcrt/5HTT double knockout (DKO) mice. Hcrt deficient mice lacking one copy of the 5HTT gene ($5HTT^{+/KO};Hcrt^{KO/KO}$) were included in our comparative analysis because removing a single allele of 5HTT was reported to affect the mouse behavior[23,24]. Our standard sleep/wake phenotyping protocol[25] (Supplementary Fig. 1a), with video monitoring to investigate cataplexy, was used to assess the distribution of vigilance states. In baseline (BL) conditions, all genotypes displayed the typical rhythmic diurnal behavior with a higher amount of wakefulness in the dark and a higher amount of NREMS during the light period. They all responded to sleep deprivation (SD) with increases in NREMS and REMS amounts (Fig. 1a, b and Supplementary Fig. 1b). The distribution of wakefulness and NREMS in all genotypes was normal and the appearance of one state occurs at the expense of the other. Prominent differences were, however, observed in REMS. REMS amount was significantly larger in genotypes with 5HTT null compared to other genotypes, most strikingly in BL light period which is the normal resting period for mice ($5HTT^{KO/KO}$: $7.14 \pm 1.2$ min; DKO: $6.71 \pm 0.59$ min vs WT: $4.71 \pm 0.47$ min; $Hcrt^{KO/KO}$: $4.96 \pm 0.53$ min; $5HTT^{+/KO};Hcrt^{KO/KO}$: $5.61 \pm 0.66$ min, mean $\pm$ SD, Fig. 1a bottom and 1b). In contrast, the increase in REMS during the dark period was found to be more pronounced in mice lacking an intact HCRT system ($Hcrt^{KO/KO}$: $3.38 \pm 0.66$ min; $5HTT^{+/KO};Hcrt^{KO/KO}$: $3.14 \pm 0.36$ min; DKO: $4.88 \pm 0.35$ min vs WT: $1.83 \pm 0.21$ min and $5HTT^{KO/KO}$: $2.04 \pm 0.5$ min, mean $\pm$ SD, Fig. 1b), a finding in accordance with previous reports[10,26]. The number of long REMS bouts (>2 min) during the dark period was, however, highest in $5HTT^{KO/KO}$ and DKO mice (Fig. 1c), suggesting a further stabilization of REMS in the absence of 5HTT. To test whether REMS stabilization results from an inability of $5HTT^{KO/KO}$ and DKO mice to terminate REMS bouts, or to switch states, we analyzed transitions into and out of REMS. This analysis (Supplementary Fig. 1c) indicated no difference in transition numbers between $5HTT^{KO/KO}$ and WT mice, while DKO mice showed significantly more transitions both in and out of REMS than WT mice. Therefore, the increase in the amount of REMS, as well as in REMS bout length in these mutant mice does not seem to be due to a defect in terminating REMS.

When the animals were homeostatically challenged with SD, only WT mice demonstrated full dissipation of their REMS debt in recovery (Rec) dark period, while REMS time remained higher in the other four genotypes, with the highest level in the DKO group (DKO: $6.33 \pm 0.41$ min vs WT: $2.4 \pm 0.46$ min; $5HTT^{KO/KO}$: $3.66 \pm 1.2$ min; $Hcrt^{KO/KO}$: $3.73 \pm 0.63$ min; $5HTT^{+/KO};Hcrt^{KO/KO}$: $3.94 \pm 0.27$ min, mean $\pm$ SD, Fig. 1b). HCRT and 5HT systems had similar effects on REMS consolidation during the dark period (both independently upregulate REMS during BL and recovery), as DKO mice exhibit a marked increase in REMS amount compared to both $Hcrt^{KO/KO}$ and $5HTT^{KO/KO}$ mice. As mentioned

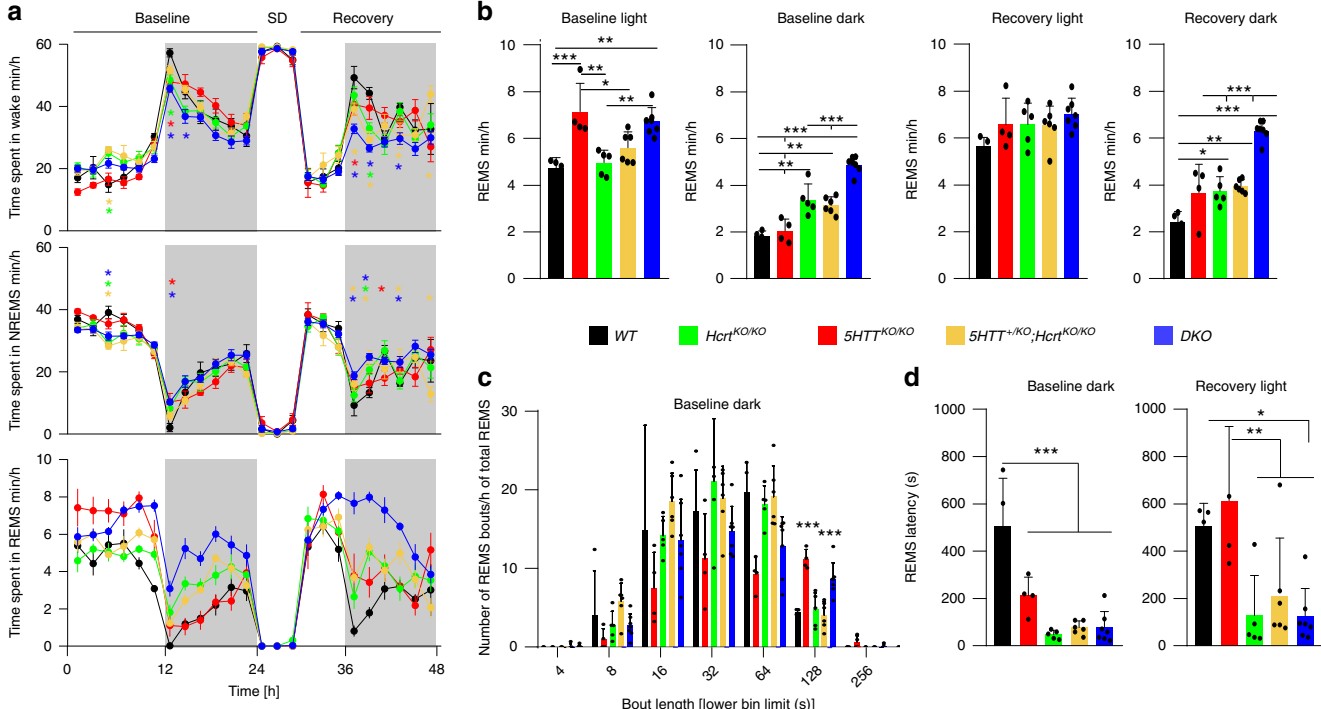

**Fig. 1 Increased REMS time in 5HTT^ko/ko mice and its regulation by HCRT. a** Time-course of vigilance states; Wakefulness (top), NREMS (middle) and REMS (bottom). *5HTT^KO/KO* mice show a large increase in REMS amount during the light period, while *Hcrt^KO/KO* and *DKO* mice show a large increase in the dark period both at baseline and during recovery (two-way ANOVA, time × genotype $F_{(92,483)} = 2.94$ (wakefulness), 2.37 (NREMS) and 4.64 (REMS), $P < 0.0001$, followed by Dunnett's test, *WT* vs *5HTT^+/KO;Hcrt^KO/KO* *, *DKO* *, *5HTT^KO/KO* *, *Hcrt^KO/KO* *, $P < 0.05$). Data points are shown in minutes per hour and represent the average of 2 h. Baseline represents the average of the 2 recording days that preceded sleep deprivation (SD). **b** From left to right: the total amount of REMS during baseline light, dark and recovery light and dark periods (one-way ANOVA, genotype $F_{(4,21)} = 10.6$, 41.80, 1.89, and 30.70, respectively, $P = 0.14$ for recovery light and $P < 0.0001$ for the rest, followed by Tukey test). **c** Distribution of REMS bout durations indicates significantly longer bouts in *5HTT^KO/KO* and DKO mice. Lower numbers for each bout duration bin are presented on the x axis (one-way ANOVA, genotype $F_{(4, 21)} = 15.03$, $P < 0.0001$, followed by Tukey test). **d** REMS latency is dramatically decreased in genotypes carrying a mutant *Hcrt* allele, both in baseline dark period and immediately after SD (one-way ANOVA, genotype $F_{(4, 21)} = 19.74$ (baseline dark, $P < 0.0001$) and 6.31 (recovery light, $P < 0.002$) followed by Tukey test). *$P < 0.05$; **$P < 0.01$; ***$P < 0.001$ for (**b–d**). Values are mean ± SD. DKO: $n = 7$, 5HTT^KO/KO: $n = 4$, Hcrt^KO/KO: $n = 5$, 5HTT^+/KO;Hcrt^KO/KO: $n = 6$ and WT: $n = 4$.

above, mice lacking the *5HTT* gene (*5HTT^KO/KO* and *DKO* mice) exhibit higher amounts of REMS during the BL light period.

To determine if any of these mutations facilitate transitions from NREMS to REMS we calculated REMS latency across all genotypes. In BL, HCRT deficient mice exhibited reduced REMS latency following sustained waking in early BL dark period. Our data additionally revealed that the three other mutant groups, including mice lacking only 5HTT, also had reduced REMS latency (*Hcrt^KO/KO*: 47.2 ± 8.13 s; *5HTT^+/KO;Hcrt^KO/KO*: 76.3 ± 11.56 s; *DKO*: 77.7 ± 25.3 s vs *WT*: 507 ± 100.4 s and *5HTT^KO/KO*: 212 ± 39.2 s, mean ± SEM, Fig. 1d, left). However, the effect seemed less robust than for *Hcrt^KO/KO* mice, as in recovery period after SD, *5HTT^KO/KO* did not differ from *WT* mice (*Hcrt^KO/KO*: 127.2 ± 75.95 s; *5HTT^+/KO;Hcrt^KO/KO*: 224 ± 94.6 s; *DKO*: 125.7 ± 44.2 sec vs *WT*: 506 ± 48.48 s and *5HTT^KO/KO*: 613 ± 156.9 s, mean ± SEM, Fig. 1d, right). Hence, under BL conditions, *5HTT^KO/KO* mice showed an increased total amount of REMS, early REMS entry, and longer REMS bouts. 5HTT loss appears to primarily foster REMS maintenance and modestly initiation. Altogether, these data indicate that 5HTT loss during the light period, whether in the presence (*5HTT^KO/KO*) or absence (*DKO*) of HCRT signaling, and HCRT loss during the dark period, whether in normal (*Hcrt^KO/KO*) or altered 5HT signaling (*5HTT^+/KO;Hcrt^KO/KO* and *DKO*), lead to an increase in REMS propensity, suggesting an important role of the 5HT and HCRT systems in REMS regulation.

**Spontaneous cataplexy is reduced in mice lacking both *Hcrt* and *5HTT* genes.** Cataplexy is elegantly recapitulated in mice lacking a functional HCRT system[10]. EEG/EMG traces of a typical cataplexy bout in a *Hcrt^KO/KO* are depicted in Fig. 2a and cataplexy characteristic spectral profile is shown in Fig. 2b. As expected, *5HTT^KO/KO* and *WT* mice displayed no cataplexies. Video recording accompanied by EEG signal analysis showed that, while *Hcrt^KO/KO* and *5HTT^+/KO;Hcrt^KO/KO* mice exhibited a high number of cataplexies both in BL and recovery dark periods (*Hcrt^KO/KO*, BL: 12.4 ± 3.3; recovery: 9.8 ± 1.5; *5HTT^+/KO;Hcrt^KO/KO*, BL: 10.25 ± 1.34; recovery: 6.66 ± 1.7, mean ± SD, Fig. 2c upper panels), *DKO* mice showed a dramatic decrease (BL: 3.21 ± 0.66; recovery: 0.57 ± 0.36). Average number of cataplexy bouts of less than 2 min in the BL dark period was also reduced in *DKO* mice as compared to *Hcrt^KO/KO* and *5HTT^+/KO;Hcrt^KO/KO* mice (Fig. 2c bottom).

As mentioned above, REMS latency is highly reduced in HCRT deficient and *DKO* mice despite a pronounced reduction of cataplexy attacks in the latter. We next investigated other narcolepsy traits such as the stability of wakefulness and NREMS (bout number and duration), and NREMS consolidation. Analysis of the quality of wakefulness in all genotypes showed that HCRT deficient mice exhibited, as expected, an increased number of waking bouts in BL dark period (BL: *Hcrt^KO/KO*: 17.2 ± 2.2; *5HTT^+/KO;Hcrt^KO/KO*: 16.49 ± 2.75; *DKO*: 17.7 ± 0.9 vs *WT*: 8.3 ± 2.42 and *5HTT^KO/KO*: 8.9 ± 1, mean ± SD, Fig. 2d right), and these

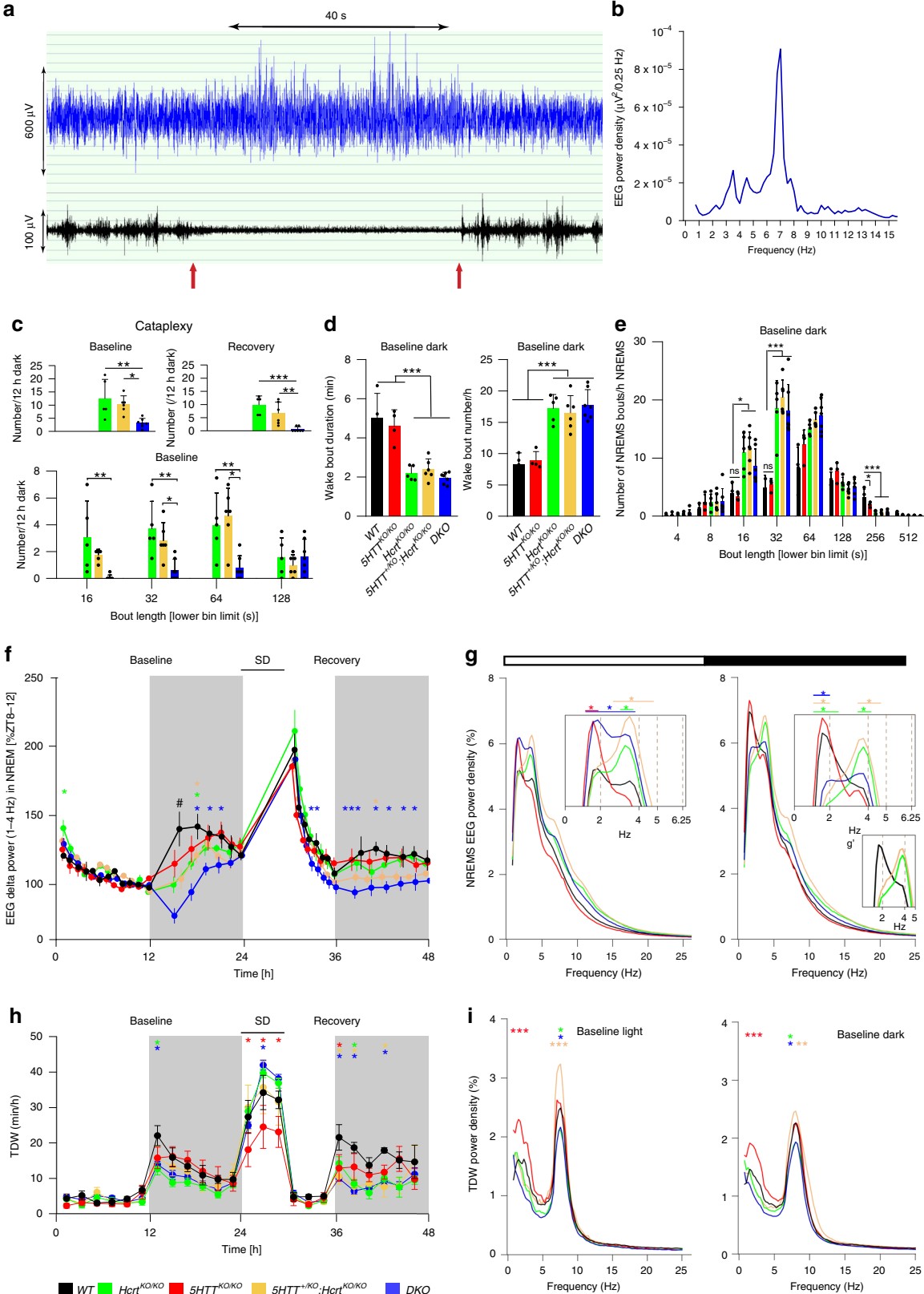

waking episodes were of shorter duration (BL: $Hcrt^{KO/KO}$: 2.2 ± 0.37 min; $5HTT^{+/KO}$;$Hcrt^{KO/KO}$: 2.39 ± 0.53 min; DKO: 1.95 ± 0.28 min vs WT: 5.00 ± 1.25 min and $5HTT^{KO/KO}$: 4.59 ± 0.83 min, mean ± SD, Fig. 2d left). All three groups with HCRT deficiency showed also a large number of shorter NREMS bouts (Fig. 2e) confirming a fragmentation of NREMS. Altogether, these data

suggest that, despite a significant reduction in cataplexy number and duration, other narcolepsy with cataplexy-related traits (such as state instability) are not affected by the introduction of null $5HTT$ alleles in $Hcrt^{KO/KO}$ background. This finding also suggests that the regulation of cataplexy in narcoleptic mice is 5HT-dependent while some other narcolepsy traits are not.

**Fig. 2 Narcolepsy symptoms and vigilance states are differentially regulated. a** EEG/EMG traces of a typical cataplexy bout in $Hcrt^{KO/KO}$ mouse and **b** the absolute power density for the 40 s EEG signal (10 × 4s epochs) indicated in (**a**). **c** Cataplexy number and duration during both baseline and recovery dark periods (one-way ANOVA, genotype $F(4, 21) = 11.72$ (baseline) and 14.99 (recovery), $P < 0.0001$, followed by Tukey test). **d** Wake bout duration (left) and number (right) in the baseline dark period. **e** Distribution of NREMS bout durations during baseline dark period (one-way ANOVA, genotype F (4, 21) = 9.7 (16 s, $P < 0.001$), 20.87 (32 s, $P < 0.0001$) and 18.12 (256 s, $P < 0.0001$), followed by Tukey test, $*P < 0.05$; $**P < 0.01$; $***P < 0.001$). Values are mean ± SD. **f** Time-course of the EEG delta power (1–4 Hz) in all genotypes in the course of the 3-day recording (baselines are averaged). Each dot represents one percentile (see "Methods") (two-way ANOVA, interaction time × genotype $F(124, 651) = 2.58$, $P < 0.0001$, followed by Dunnett's test, $WT$ vs $5HTT^{KO/KO}$ *, $5HTT^{+/KO};Hcrt^{KO/KO}$ *, $DKO$ *, $Hcrt^{KO/KO}$ *, $P < 0.05$, # $P < 0.01$ WT vs all other genotypes). **g** EEG power spectra of all genotypes in NREMS during light (left) and dark (right) periods. Insets indicate amplification for specific EEG frequencies and **g'** represents the lack of slow-delta activity in cataplexy-exhibiting mice (two-way ANOVA, interaction frequency × genotype $F(388, 2037) = 3.244.8$ (light), 2.52 (dark), $P < 0.0001$, followed by Dunnett's test, $WT$ vs $5HTT^{+/KO};Hcrt^{KO/KO}$ *, $DKO$ *, $5HTT^{KO/KO}$ *, $Hcrt^{KO/KO}$ * $P < 0.05$). Values are mean ± SD. **h** Time-course distribution of TDW amounts. Data points are shown in minutes/hour and represent the average of 2 h (baselines are averaged), (two-way ANOVA, interaction time × genotype $F(92, 483) = 2.63$, $P < 0.0001$, followed by Dunnett's test, $WT$ vs $5HTT^{+/KO};Hcrt^{KO/KO}$ *, $DKO$ *, $5HTT^{KO/KO}$ *, $Hcrt^{KO/KO}$ * $P < 0.05$). **i** EEG power spectra in TDW during light (left) and dark (right) periods. Values are mean ± SD and expressed as the percentage of each mouse's total baseline EEG power (two-way ANOVA, interaction frequency × genotype $F(388, 2037) = 3.37$ (light) and 1.84 (dark), $P < 0.0001$, followed by Dunnett's test, $*P < 0.05$; $**P < 0.01$; $***P < 0.001$, $WT$ vs $5HTT^{+/KO};Hcrt^{KO/KO}$ *, $DKO$ *, $5HTT^{KO/KO}$ *, $Hcrt^{KO/KO}$ *. $DKO$: n = 7, $5HTT^{KO/KO}$: n = 4, $Hcrt^{KO/KO}$: n = 5, $5HTT^{+/KO}$; $Hcrt^{KO/KO}$: n = 6 and $WT$: n = 4.

**Impact of 5HTT deficiency on NREMS need in mice lacking the *Hcrt* gene**. To gain insight into the homeostatic regulation of sleep among mice genotypes, we analyzed the time-course of the NREMS EEG delta power (1–4 Hz) across our 3-day recordings. All genotypes showed a typical decline in delta power during the light (resting) period and a strong rebound after SD, suggesting an intact homeostatic regulation of sleep need (Fig. 2f). However, the build-up in delta power in the BL early dark period was greatly attenuated in all genotypes compared to $WT$ mice, especially in $DKO$ mice (Fig. 2f).

The deficit in delta power in the four mutant groups was independent of the time spent in NREMS as there were no marked differences in NREMS amount between genotypes in BL dark period (Fig. 1a, middle). This indicates that there is a deficient build-up in homeostatic sleep need during the major waking period. In this regard, $5HTT^{KO/KO}$ mice were the closest to $WT$ mice and recovered their build-up deficiency earlier than did the other mutant animals. NREMS spectral analysis during the BL light period revealed other critical differences. $5HTT^{KO/KO}$ and $DKO$ mice spent more time in REMS during the light period and expressed higher slow-delta (1–2 Hz) activity relative to $WT$, while fast-delta (3–4 Hz) frequencies were mainly increased in $5HTT^{+/KO};Hcrt^{KO/KO}$, $Hcrt^{KO/KO}$ and $DKO$ mice (Fig. 2g, left). $DKO$ mice hence show both slow- and fast-delta responses, indicating cumulative and probably independent effects of HCRT and 5HT systems in regulating NREMS need, similar to their effects on REMS amount. In BL dark period, NREMS of $5HTT^{KO/KO}$ mice resembled $WT$ animals in showing a robust slow-delta activity, while $Hcrt^{KO/KO}$ and $5HTT^{+/KO};Hcrt^{KO/KO}$ mice exhibited a marked blunting in this frequency range (Fig. 2g right, g'). Blunting of slow-delta activity was previously reported to characterize the NREMS of $Hcrt^{KO/KO}$ and $nNOS^{KO/KO}$ mice in BL, and of noradrenergic cell-specific $Hcrtr1$ conditional knock-out mice following periods of enhanced arousal[26–28]. This response in NREMS spectral characteristics was hypothesized to reflect deficits in neural plasticity processes in prior wakefulness[29], and removal of one $5HTT$ allele ($5HTT^{+/KO};Hcrt^{KO/KO}$ mice) does not affect this phenotype. NREMS of $DKO$ mice during the dark period appears extremely shallow, exhibiting the cumulative effects of a deficiency in slow-delta power as in $Hcrt^{KO/KO}$ mice, and a deficiency in fast-delta power as in $5HTT^{KO/KO}$ mice. Collectively, it can be concluded that the precise regulation of fast and slow-delta power as proxy for the sleep homeostatic process, is highly dependent on HCRT and 5HT systems.

**Differential expression of theta-dominated wakefulness (TDW) in cataplexy-exhibiting genotypes**. Some portion of wakefulness is associated with goal-oriented, exploratory and motivated behaviors, such as nest building, running, or drinking, and often referred to in rodents as "active wakefulness". These behaviors act as cataplexy triggers in mice with HCRT deficiency, and share the EEG signature called "theta-dominated wakefulness" (TDW 6.0–9.5 Hz), which is also the principal driver of sleep homeostat[26]. During the BL early dark period, $Hcrt^{KO/KO}$ and $DKO$ mice showed significantly reduced amount of TDW as compared to their $WT$ littermates, while $5HTT^{KO/KO}$, and $5HTT^{+/KO};Hcrt^{KO/KO}$ mice did not (Fig. 2h). Since all these mutant genotypes displayed a deficit in building-up of the homeostatic sleep need in the early dark period (Fig. 2f), it can be inferred that at least for $5HTT^{+/KO};Hcrt^{KO/KO}$ and $5HTT^{KO/KO}$ mouse lines, TDW amount does not drive their altered response to homeostatic sleep need. Although they displayed a normal amount of TDW in BL, $5HTT^{KO/KO}$ mice exhibited a markedly diminished TDW expression during enforced wakefulness. This indicates that normal 5HT transmission is necessary to express high-quality wakefulness in conditions of heightened arousal such as during SD. An interesting and paradoxical finding here is that the deficit of $5HTT^{KO/KO}$ mice in exhibiting normal active wakefulness during SD was HCRT-dependent, as absence of HCRT fully normalized time spent in TDW during SD in $DKO$ mice. To next assess the quality of our mutants' TDW states, spectral profiles were compared. This revealed that, during both light and dark periods, $5HTT^{KO/KO}$ mice expressed a TDW state with EEG theta power similar to one of $WT$ mice, whereas $5HTT^{+/KO};Hcrt^{KO/KO}$ mice displayed higher, and $Hcrt^{KO/KO}$ and $DKO$ mice lower, theta power values (Fig. 2i). Hence the quality of wakefulness, at least as indexed by TDW theta power, is not necessarily linked to the incidence of cataplexy since both $Hcrt^{KO/KO}$ and $5HTT^{+/KO};Hcrt^{KO/KO}$ mice exhibit almost the same number of cataplexy attacks (Fig. 2c).

**REMS and waking EEG power densities are modulated by 5HT**. As shown above, $5HTT^{KO/KO}$ and $DKO$ mice exhibit heightened REMS amount during the light period while increased REMS in the dark period characterizes $Hcrt^{KO/KO}$, $5HTT^{+/KO};Hcrt^{KO/KO}$ and $DKO$ mice. We next analyzed BL REMS spectra in the light and dark periods across all genotypes. A large increase in theta power was observed in $5HTT^{+/KO};Hcrt^{KO/KO}$ mice compared to all other genotypes (Fig. 3a). To rule out an effect due to our

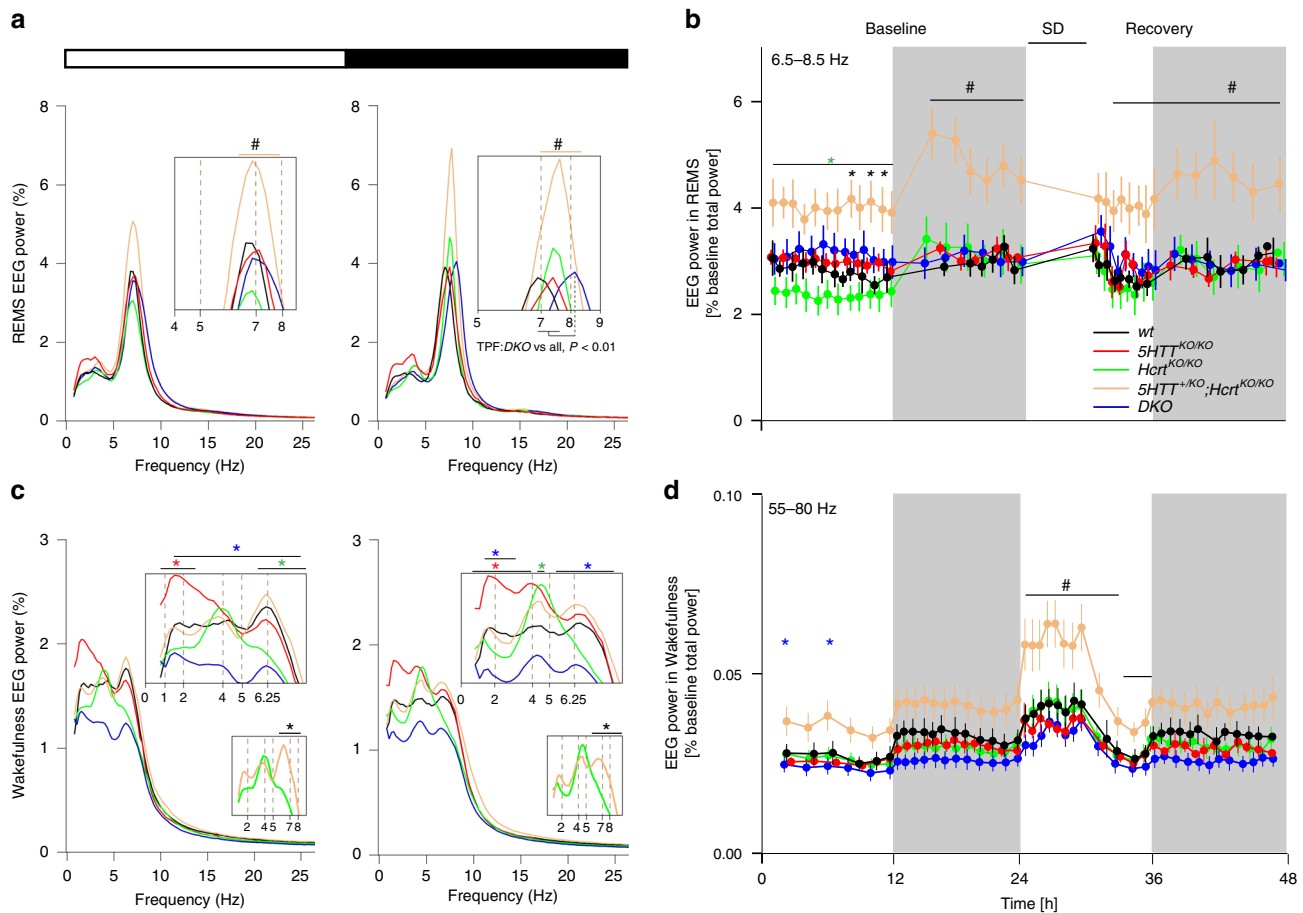

**Fig. 3 REMS EEG theta power is increased in *5HTT$^{+/KO}$;Hcrt$^{KO/KO}$* mice. a** EEG power spectra in baseline REMS during light (left) and dark (right) periods. (two-way ANOVA, interaction frequency × genotype $F$ (388, 2037) = 3.34 (light) and 6.35 (dark), $P < 0.0001$, followed by Dunnett's test, *5HTT$^{+/KO}$;Hcrt$^{KO/KO}$* vs all # $P < 0.05$)· Insets show higher resolution. REMS theta peak frequency (TPF) is faster in *DKO* mice compared to all other genotypes. **b** REMS theta (6.5–8.5 Hz) power dynamics in the course of the 3-day recording. Powers within this frequency range are higher during the dark period (baseline one and two are averaged) (two-way ANOVA, interaction time × genotype $F$ (124, 651) = 2.97, $P < 0.0001$, followed by Dunnett's test, *5HTT$^{+/KO}$;Hcrt$^{KO/KO}$* vs WT *, DKO*, *5HTT$^{KO/KO}$* *, *Hcrt$^{KO/KO}$**, all $P < 0.05$). **c** Waking baseline EEG power spectra for all genotypes during baseline light (left) and dark (right) periods. The values are expressed as the percentage of each mouse's total baseline EEG power. Insets show higher resolution. Smaller insets represent differential expression of waking theta power in cataplexy-exhibiting mice (two-way ANOVA, interaction frequency × genotype $F$ (388, 2037) = 3.04 (light) and 2.28 (dark), $P < 0.0001$, followed by Dunnett's test, WT vs *5HTT$^{+/KO}$;Hcrt$^{KO/KO}$* *, *DKO* *, *5HTT$^{KO/KO}$* *, *Hcrt$^{KO/KO}$* * and # $P < 0.05$, insets *5HTT$^{+/KO}$;Hcrt$^{KO/KO}$* vs *Hcrt$^{KO/KO}$* * $P < 0.05$). **d** Waking EEG fast-gamma (55–80 Hz) power dynamics in the course of the 3-day recording (baseline one and two are averaged) (two-way ANOVA, interaction time × genotype $F$ (164, 861) = 3.8, $P < 0.0001$, followed by Dunnett's test, *5HTT$^{+/KO}$;Hcrt$^{KO/KO}$* vs WT *, DKO*, *5HTT$^{KO/KO}$* *, *Hcrt$^{KO/KO}$** $P < 0.05$; **$P < 0.01$; ***$P < 0.001$). # in (**a, b**) and (**d**) $P < 0.05$, *5HTT$^{+/KO}$;Hcrt$^{KO/KO}$* vs all other genotypes. Values are mean ± SD. *DKO*: $n = 7$, *5HTT$^{KO/KO}$*: $n = 4$, *Hcrt$^{KO/KO}$*: $n = 5$, *5HTT$^{+/KO}$;Hcrt$^{KO/KO}$*: $n = 6$ and WT: $n = 4$.

relative EEG power measures, we analyzed the absolute EEG spectral power in REMS across all genotypes, and found very similar results (Supplementary Fig. 2a). This indicates a significant contrast between the two cataplexy-expressing genotypes as *Hcrt$^{KO/KO}$* mice exhibited a significantly lower REMS power density within the theta peak (6–8 Hz) frequencies (TPF). In addition, while the REMS TPF of WT, *Hcrt$^{KO/KO}$*, *5HTT$^{+/KO}$; Hcrt$^{KO/KO}$*, and *5HTT$^{KO/KO}$* mice overlapped, the peak frequency of *DKO* mice during the dark period was significantly faster (TPF; *DKO*: 8.25 ± 0.2 Hz vs WT: 7.18 ± 0.31 Hz, *5HTT$^{KO/KO}$*: 7.43 ± 0.12 Hz, *Hcrt$^{KO/KO}$*: 7.55 ± 0.20 Hz and *5HTT$^{+/KO}$;Hcrt$^{KO/KO}$*: 7.62 ± 0.34 Hz, mean±SD, Fig. 3a, right).

To delineate if the heightened REMS theta power in *5HTT$^{+/KO}$; Hcrt$^{KO/KO}$* mice corresponds to specific time points, the dynamics of the REMS EEG theta power (6.5–8.5 Hz) was analyzed across the three recording days (Fig. 3b). *5HTT$^{+/KO}$;Hcrt$^{KO/KO}$* mice displayed increased theta power throughout the recording days, which was particularly pronounced during the dark periods and

immediately after SD (Fig. 3b). This finding is not due to differences in REMS amounts as *5HTT$^{+/KO}$;Hcrt$^{KO/KO}$* and *Hcrt$^{KO/KO}$* mice express similar amounts of REMS in BL and after SD (Fig. 1b).

To determine whether changes in NREMS and REMS spectral profiles can be traced back to any specific prior waking activities, we analyzed the BL waking EEG spectral properties and dynamics. Spectral analysis of wakefulness lower frequency bands (0.75–25 Hz) showed that *5HTT$^{KO/KO}$* mice, which spent more time in REMS, displayed a pronounced increase in waking slow-delta power (0.75–2.5 Hz) during the light period, and across the entire delta range (0.75–4 Hz) during the dark period, as compared to WT mice (Fig. 3c). *DKO* mice exhibited a pronounced blunting in the 1.5–7 Hz frequency range during the light, and 1.5–3.25 and 5.25–8.25 Hz during the dark period, as compared to WT mice. Waking state comparisons between *5HTT$^{+/KO}$;Hcrt$^{KO/KO}$* and *Hcrt$^{KO/KO}$* genotypes revealed a significant decrease in *Hcrt$^{KO/KO}$* mice in the theta band

(6–8 Hz) during both light and dark periods (Fig. 3c, insets). This difference is not linked to cataplexy incidence as the two lines exhibit similar cataplexy number and duration. Blunting of waking theta (6–8 Hz) activity in $Hcrt^{KO/KO}$ mice and the prevailing peak in the frequency band between 4–5 Hz were recently reported[26]. Our present findings suggest that the addition of a $5HTT$ null allele attenuates this deficit in BL waking theta activity of $Hcrt^{KO/KO}$ mice.

When waking higher frequency bands were dynamically examined across the 3 experimental days, $5HTT^{+/KO};Hcrt^{KO/KO}$ mice exhibited an increased gamma power (55–80 Hz) compared to all other genotypes, which became more apparent during SD and the following recovery light period (Fig. 3d). This increase is not seen in other waking frequency bands (1–2, 6–8 and 32–45 Hz) (Supplementary Fig. 2b, c, d). As shown above (Fig. 2i), $5HTT^{+/KO};Hcrt^{KO/KO}$ mice also exhibit a considerably higher theta power during TDW episodes. Note also that genotypes with more REMS ($5HTT^{KO/KO}$ and $DKO$) have a less powerful slow-gamma (32-45 Hz) activity (Supplementary Fig. 2c). Altogether, these results suggest that mechanisms underlying REMS latency ($Hcrt^{KO/KO}$ mice; shorter REM latency), REMS duration ($5HTT^{KO/KO}$ mice; longer REM duration) and REMS theta power ($5HTT^{+/KO};Hcrt^{KO/KO}$ mice; high REMS theta power) are differentially regulated.

**Disruption of HCRT input into the dorsal raphe consolidates REMS and induces large changes in the EEG spectral profiles across all states.** Our data suggest that increased extracellular 5HT leads to near suppression of cataplexy attacks in HCRT deficient mice and increased REMS. Restoring HCRT receptors 1&2 in the DR of mice lacking the two receptors was reported to result in a complete suppression of cataplexy[20]. These findings suggest a strong interaction between HCRT and 5HT systems in regulating normal vigilance states and cataplexy. Therefore, we sought to determine if the removal of HCRT receptors 1&2 from DR neurons of adult $WT$ mice (Fig. 4a) affects the regulations of cataplexy and REMS. To this end, we transduced DR neurons of mice harboring $loxP$ site-flanked alleles of both HCRT receptors 1&2 genes[27] with AAV expressing $mCherry$ and Cre recombinase. In these mice, Cre recombination results in the removal of both HCRT receptors and their replacement by expression of the GFP reporter[27]. Mice with a substantial fraction of transduced DR cells were chosen for sleep/wake state scoring and EEG analysis. Immunostaining of coronal brain sections and cell counting across the DR region revealed substantial co-localization of the 5HT cell-specific marker Tryptophan Hydroxylase (TPH) with mCHERRY ($77.85\% \pm 2.95\%$, mean $\pm$ SD, $n = 4$; Fig. 4b). We also found that $81.64 \pm 5.21\%$ (mean $\pm$ SD, $n = 4$) of mCherry-positive cells were TPH-positive. We verified that mCherry-immunoreactivity largely reflected Cre recombinase expression ($73.82 \pm 1.64\%$, (mean $\pm$ SD, $n = 3$) of mCherry-positive neurons were CRE-positive while $77.60 \pm 3.62\%$ of CRE-positive cells were mCherry-positive) (Supplementary Fig. 3). Moreover, we found strong GFP expression in mCHERRY-positive neurons, indicating that recombination had occurred (Fig. 4c, d). Control animals ($Hcrtr1^{flox/flox};Hcrtr2^{flox/flox}$ mice) were injected with a $mCherry$-expressing viral vector lacking Cre recombinase.

Vigilance states analysis showed a normal distribution, with more wakefulness during the dark and NREMS during the light period in both genotypes. No substantial differences between groups were observed in the amount of sleep and wakefulness during BL and after SD (Supplementary Fig. 5a). Careful video and EEG analysis failed to detect any cataplexy attack in $DR\_Hcrtr1\&2\text{-}cKO$ mice indicating that acute disruption of HCRT signaling in the DR is insufficient to induce cataplexy in

adult $WT$ animals. However, we observed a prominent consolidation of REMS and a fragmentation of NREMS predominantly during the dark period in $DR\_Hcrtr1\&2\text{-}cKO$ mice, as compared to control animals (Fig. 5a, b). This indicates that an intact $LH^{HCRT} - DR^{5HT}$ circuit is necessary to limit sleep fragmentation and improve REMS consolidation during the dark period.

EEG spectral analysis indicated significant differences between control and $DR\_Hcrtr1\&2\text{-}cKO$ mice. During both light and dark periods, $DR\_Hcrtr1\&2\text{-}cKO$ mice showed increased slow-delta (0.75–1 Hz) power in BL wakefulness (Fig. 5c and insets) which was also found more pronounced in both light and dark periods after SD (Supplementary Fig 4a). We also observed a large increase in slow-delta power during the light (0.75–2.5 Hz) and dark periods (1.25–2.25 Hz) of the BL NREMS in $DR\_Hcrtr1\&2\text{-}cKO$ mice, as compared to control mice (Fig. 5d). SD challenge resulted in the typical homeostatic response of this frequency band, with a large rebound observed immediately after SD (Supplementary Fig. 4b and Supplementary Fig. 5b). The increase in slow-delta power in $DR\_Hcrtr1\&2\text{-}cKO$ mice is in contrast to the blunting of this activity in $Hcrt^{KO/KO}$ mice (Fig. 5f), as also previously reported[26]. A higher delta power during wakefulness was previously shown to be linked to high sleep pressure and decreased vigilance in humans[30] and mice[27]. These data suggest that removing HCRT receptors from DR-5HT cells enhances sleep need both during spontaneous and enforced wakefulness.

Contrary to waking and NREMS slow-delta power, activity in the waking fast-delta and theta powers from 2 to 7.75 Hz in light period, and from 1.75 to 8.5 Hz in dark period, was largely dampened in $DR\_Hcrtr1\&2\text{-}cKO$ mice, as compared to control animals (Fig. 5c). This indicates, as reported before[26], that HCRT is necessary to sustain the theta-rich waking state. Our finding suggests that this HCRT activity is partially mediated by $LH^{HCRT}$-$DR^{5HT}$ circuits. REMS, which is highly consolidated in $DR\_Hcrtr1\&2\text{-}cKO$ mice, also exhibited a significant dampening of the theta power density during both light and dark periods (Fig. 5e). Altogether, these data suggest that HCRT input into the DR nucleus is necessary to stabilize normal vigilance states.

**$5HTT^{KO/KO}$ mice have partial HCRT deficiency.** To gain insight into neuropeptide regulation in our knockout animals we quantified the transcriptional level of wake ($Hcrt$) and sleep-promoting ($Pmch$ and $Qrfp$) genes in the hypothalamus of our five genotypes. While the lack of $Hcrt$ expression is expected in $Hcrt^{KO/KO}$ mice, a large reduction in $Hcrt$ expression was also observed in $5HTT^{KO/KO}$ mice compared to $WT$ littermates (WT: $1.00 \pm 0.11$, $5HTT^{KO/KO}$: $0.63 \pm 0.06$, arbitrary units, mean $\pm$ SEM, Fig. 6). Also, $Qrfp$ expression tended to be largely reduced in $5HTT^{KO/KO}$ mice compared to their $WT$ littermates. No significant change, however, was observed in the expression of the REMS-promoting $Pmch$ gene.

## Discussion

In this study, the interplay between the HCRT and 5HT systems in the regulation of vigilance states was investigated. The essential role of HCRT neurotransmission in the normal regulation of vigilance states, specifically waking and cataplexy, is well-established[11,31]. Also, the involvement of the 5HT system in stabilizing the normal wakefulness and REMS is documented[12,32]. HCRT neurons by expressing 5HT1A receptors[33] and DR 5HT neurons by expressing HCRT receptors 1&2[34] reciprocally interact with each other to fine-tune a broad array of physiological processes, including vigilance states[35].

We showed that $5HTT$ deletion in $Hcrt^{KO/KO}$ mice leads to a near suppression of cataplexy attacks, without affecting other

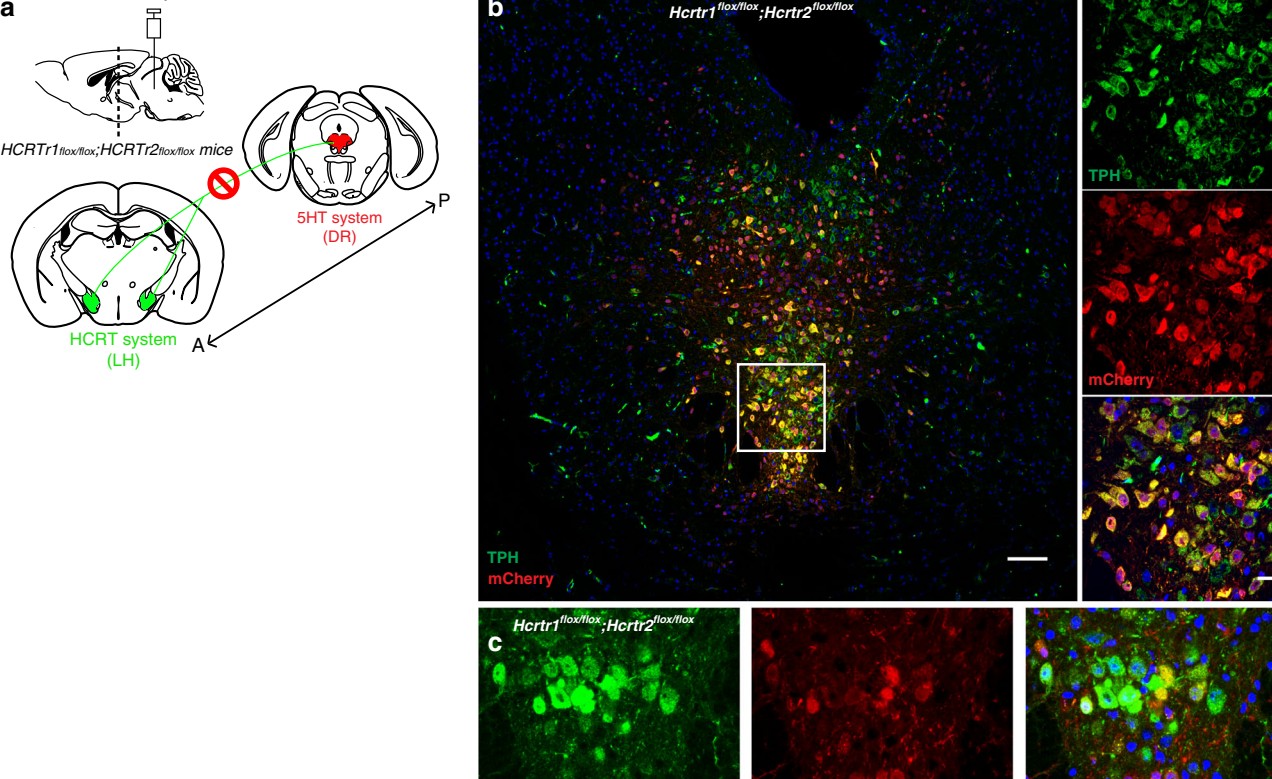

**Fig. 4 Inactivation of *Hcrtr1&2* in the DR. a** Schematic figure depicting the experiment where the HCRT input (from posterior "P" to anterior "A") to the dorsal raphe (DR) is disrupted. **b** Representative confocal image of the DR of *Hcrtr1&2 double-floxed* homozygous mice injected with AAV-EF1a-mCherry-IRES-Cre-WPRE and stained with mCHERRY (red) and TPH (green) antibodies. mCHERRY-positive cells (red) are highly co-localized with TPH-positive cells (green). Higher magnification (×40, right). **c, d** GFP is expressed after Cre recombination. TPH-positive cells (**c** middle red) transduced with Cre virus (**d** middle red) are positive for GFP (**c** and **d**, left green). Scale bars: **b** 100 μm, rest 20 μm.

features of narcolepsy such as short REMS latency or fragmented wakefulness and NREMS. This finding is consistent with the fact that 5HT reuptake inhibitors suppress cataplexy in narcolepsy patients[36–38]. In contrast to cataplexy suppression, a large increase in REMS was found in *DKO* mice, indicating a dissociation between REMS and cataplexy and suggesting that the two states are regulated by different mechanisms, as also suggested by Hasegawa et al.[20]. REMS and cataplexy are different brain states[25] that share many similarities including muscle atonia and high EEG theta activity. Accordingly, cataplexy is thought to result from inappropriate intrusion of REMS paralysis into wakefulness. These similarities lead to speculate that they might share common underlying mechanisms. *DKO* mice exhibit a largely reduced number of cataplexy episodes during the dark period (when nearly all cataplexies occur), while they spend significantly more time in REMS. We do not rule out the possibility that cataplexy and REMS share partially common pathways[39], but

our results indicate that their generation is differentially controlled. Note also that we previously reported that high-amplitude theta-frequency paroxymal events occurring during cataplexy of mice and children are, unlike REM-sleep theta, confined to the prefrontal cortex in mice[25]. Hence the 2 states' theta signatures have at least partially distinct origins. Yet both cataplexy and REMS are under dual regulation of HCRT and 5HT systems. Cataplexy is the pathognomonic symptom of HCRT deficiency, and *Hcrt*[KO/KO] mice express more REMS in dark periods, while they show profoundly diminished REMS latencies following prolonged wakefulness, as do human narcolepsy patients. We show that *5HTT*[ko/ko] mice express higher amount of REMS in BL light periods, and, although not as severely as *Hcrt*[KO/KO] mice, also show shorter REMS latencies after prolonged waking.

Supporting the role of 5HT neurotransmission in cataplexy, Hasegawa and colleagues showed that HCRT neuron-dependent activation of 5HT release in the amygdala is required to suppress

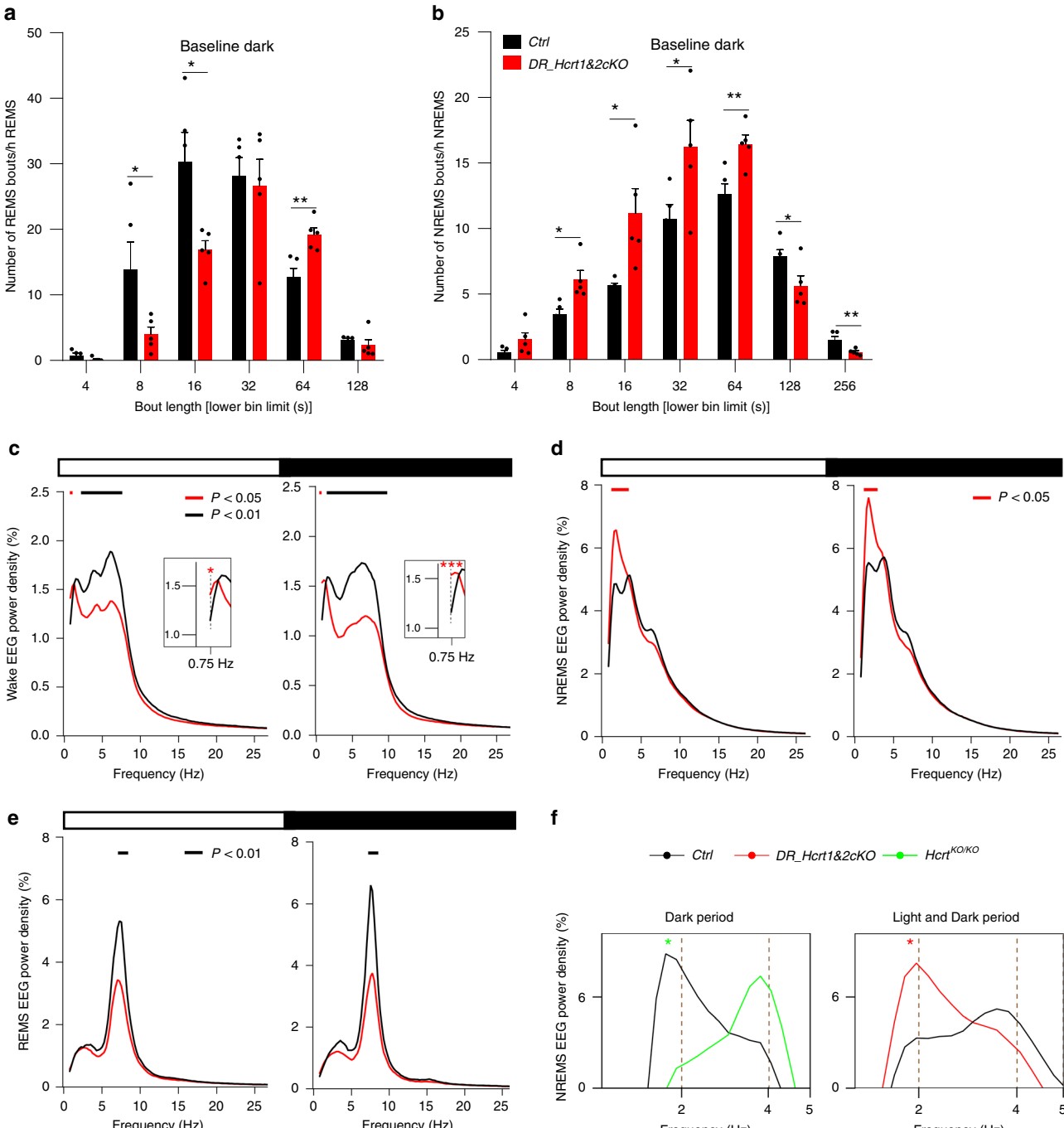

**Fig. 5 Profound alterations in vigilance states quality in DR *Hcrtr1&2* gene-inactivated mice. a** Distribution in REMS bout durations during the dark period. Lower numbers in each bout duration bin are presented on the *x* axis. REMS is markedly consolidated in *DR_Hcrtr1&2-cKO* mice (unpaired 2-tailed *t*-test, *t* (8) = 2.27 for 8 s, *P* = 0.05, 2.88 for 16 s, *P* < 0.05, and 3.87 for 64 s, *P* < 0.01, mean ± SEM). **b** Distribution of NREM bout durations during the dark period. NREMS is highly fragmented in *DR_Hcrtr1&2-cKO* mice (unpaired two-tailed *t*-test, *t* (8) = 3.25 for 8 s, *P* < 0.05, 2.86 for 16 s, *P* < 0.05), 2.36 for 32 s, *P* < 0.05, 3.61 for 64 s, *P* < 0.01, 2.41 for 128 s, *P* < 0.05) and 3.39 for 256 s, *P* < 0.01, mean ± SEM). **c** Baseline waking EEG power spectra during light (left) and dark (right) periods. Data are normalized to the total EEG power during baseline. Insets show higher magnification of lower frequencies (two-way ANOVA, interaction frequency × genotype *F* (196, 1568) = 3.86 (light), 5.7 (dark), *P* < 0.0001, followed by Dunnett's test, horizontal bars join frequency bins for which *P* < 0.05). **d** Baseline EEG power spectra for NREMS (two-way ANOVA, interaction frequency × genotype *F* (196, 1568) = 2.26 (light), 1.81 (dark), *P* < 0.0001, followed by Dunnett's test, *P* < 0.05). **e** Baseline EEG power spectra for REMS (two-way ANOVA, interaction frequency × genotype *F* (196, 1568) = 6.51 (light), 7.15 (dark), *P* < 0.0001, followed by Dunnett's test, *P* < 0.05). **f** The slow-delta in NREMS of *Hcrt^KO/KO^* and *DR_Hcrtr1&2-cKO* mice. *Hcrt^KO/KO^* mice show a profound decrease, while the disruption of HCRT input to the DR largely enhances NREMS slow-delta power (two-way ANOVA, interaction frequency × genotype *F* (388, 2037) = 2.52 (left), *F* (196, 1568) = 2.26 (right), *P* < 0.0001, followed by Dunnett's test, *WT* vs. *Hcrt^KO/KO^** and vs. *DR_Hcrtr1&2-cKO** *P* < 0.05). *DR_Hcrtr1&2-cKO* *n* = 5, and control mice *n* = 5.

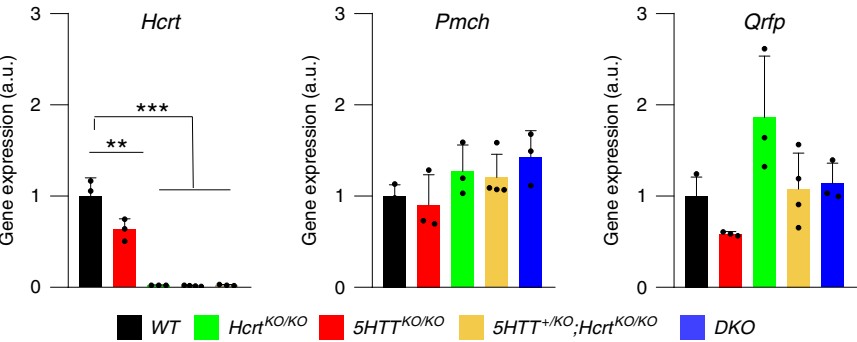

**Fig. 6 Inactivation of *5HTT* gene leads to partial HCRT deficiency.** qPCR quantification of hypothalamic RNA. *Hcrt* is largely downregulated in *5HTT^{KO/KO}* mice as compared to *WT* littermates. Values are mean ± SEM, $n = 3$ (one-way ANOVA, genotype $F_{(4, 11)} = 64.79$, $P < 0.0001$, followed by Dunnett's test, **$P < 0.01$; ***$P < 0.001$). DKO: $n = 3$, *5HTT^{KO/KO}*: $n = 3$, *Hcrt^{KO/KO}*: $n = 3$, *5HTT^{+/KO};Hcrt^{KO/KO}*: $n = 4$ and *WT*: $n = 3$.

cataplexy[40]. Our result however indicated that acute disruption of the $LH^{HCRT}$-$DR^{5HT}$ circuit is not sufficient to induce cataplexy in *WT* adult mice. This might be due to the fact that complete disruption of the $LH^{HCRT}$-$DR^{5HT}$ circuit may not have been reached in our experiment. Not all $DR^{5HT}$ cells were infected and in infected cells not all four floxed alleles may have been deleted. In addition, some cells that showed Cre expression were not TPH positive and these cells may also play a role in cataplexy (for instance local inhibitory GABA cells). A higher level of Cre recombination and the use of cell type-specific promoters are necessary in future experiments. It is however noteworthy that because 5HT acts not only at the level of synapses, but also by volume diffusion on extra-synaptic receptors, we cannot exclude that longer delays between Cre AAV injection and recording would have detected cataplexy-like events. SSRIs' anti-depressive effects are known to necessitate several weeks.

*5HTT* knockout mice were reported to have increased basal extracellular serotonin[41], which results in increased REMS (our data and others)[19]. Disruption of 5HTT function leads to major pleiotropic central and peripheral phenotypes[24], but whether any of these has a direct impact on REMS quality and quantity is unknown. *5HTT^{KO/KO}* mice display altered expression and function of 5-HT1A and 1B receptors[42] and fail to show a delayed increase in REMS under restraint stress, as do *WT* mice[43]. The latter effect was suggested to result from enhanced hypocretinergic neurotransmission[43]. Our findings however challenge this interpretation, as we found increased REMS in *5HTT^{KO/KO}* mice both in presence or absence of functional HCRT neurotransmission. It was proposed that the overexpression of REMS in adult *5HTT^{KO/KO}* mice results from excess extracellular 5HT brain levels during development, as early life blockade of 5HT synthesis, or reducing excessive 5HT1AR stimulation by endogenous 5HT, result respectively, in suppression, or limitation of REMS increase[44]. In addition, during development, 5HTT is transiently expressed in the prefrontal cortex[45,46] where 5HTT-expressing neurons (which are not serotonergic) coordinate normal prefrontal cortex-DR innervation. Since the prefrontal cortex is the site through which positive emotions trigger cataplexy[25,47], it can be speculated that developmentally modified prefrontal cortex by excess 5HT cannot trigger cataplexy.

REMS upregulation was documented in mice lacking either *Slc6a4*[19,44] or *Hcrt* genes[10,26]. We showed that *Hcrt^{KO/KO}* mice have an increase in REMS amount predominantly during the dark (active) period, relative to *WT* and *5HTT^{KO/KO}* mice. HCRT deficiency is established to result in cataplexy and sleep dysregulation including NREMS fragmentation and sleep-onset REM periods[1]. In contrast, the increase in REMS amount in *5HTT^{KO/KO}* is mainly evident during the light period, which is the physiological time for REMS expression. Recent studies showed that

intact 5HT tone is necessary to maintain REMS expression[32] and its input to HCRT neurons is necessary to maintain REMS architecture[48]. We also showed here that disruption of HCRT input to DR neurons significantly consolidates REMS. Altogether, these data indicate a strong interplay between HCRT and 5HT systems to fine-tune REMS expression and regulation.

Our data show that *5HTT^{KO/KO}* and *DKO* mice, which display more REMS during the light period, express a robust increase in slow-delta activity during NREMS. Interestingly, profound slow-delta upregulation is also seen in *DR_Hcrtr1&2-cKO* mice. This is in striking contrast with *Hcrt^{KO/KO}* and *5HTT^{+/KO};Hcrt^{KO/KO}* mice, which display more REMS in dark periods, and show a markedly reduced slow-delta power in dark period NREMS. In addition, a large increase in theta power was found in REMS of *5HTT^{+/KO};Hcrt^{KO/KO}* mice. The question remains as to how the removal of a single allele of *5HTT* in *Hcrt^{KO/KO}* mice affects so markedly REMS theta power and causes the other state-specific spectral changes seen in this genotype? The exaggerated behavior of heterozygous animals as compared to WT and KO counterparts is documented[49,50] for some genes. In addition, whether site-specific monoallelic expression of the *Slc6a4* gene underlies these findings warrants further investigations. Our data at gene expression level show that heightened 5HT tone reduces the normal expression of *Hcrt* gene, which is consistent with the inhibitory role of 5HT on HCRT neurons[33]. In *5HTT^{+/KO};Hcrt^{KO/KO}* mice with high theta power in REMS, we also observed higher powers in theta and gamma bands during wakefulness. The dependency of REMS regulation on prior wakefulness and NREMS was debated in several hypotheses[51–53]. Benington and Heller proposed a role for REMS as a homeostatic balance for NREMS[51], while, Endo et al. hypothesized that some aspects of wakefulness could functionally substitute for REMS[52]. Recently, a specific portion of the prior wakefulness was proposed to drive NREMS need[26]. Whether or not there is a link between NREMS intensity, prior wakefulness and REMS regulation also demands further investigations.

We found that *5HTT^{KO/KO}* mice spent considerably less time in the active (theta-rich) waking state when subjected to SD, but this deficit could be fully rescued by the further removal of *Hcrt* gene in *DKO* mice. Our observation thus suggests that in *5HTT^{KO/KO}* mice, excess 5HT impairs TDW state initiation or maintenance through HCRT-dependent mechanisms. This may involve the 5HT binding to 5HT1A receptors of HCRT cells and the suppression of HCRT release.

Disruption of HCRT input to the DR nucleus robustly increased slow-delta activity during NREMS, in striking contrast with *Hcrt^{KO/KO}* mice, which in BL dark period show profound NREM slow-delta wave deficiency, as do *nNOS^{KO/KO}* mice, and norepinephrine-depleted rats after SD[28,29], suggesting that this EEG frequency range is particularly susceptible to disruption.

In summary, our data highlight several important aspects of not only cataplexy and REMS regulation but also of the EEG correlates of NREM and wakefulness where reciprocal interactions between HCRT and 5HT systems play a crucial role.

## Methods

**Animals**. Two sets of mice were used in this study. To generate the first set, mice with an inactivated *Hcrt* gene[10] were mated with mice carrying the *Slc6a4*[tm1Kpl] KO allele of the gene encoding 5HTT[22]. The mice for the second set were generated by our laboratory[27,54]. *Hcrtr1/Hcrtr2* double-floxed mice are homozygous for a conditional knockout (floxed) mutation at the *Hcrtr1* locus (*Hcrtr1*[tm1.1Ava])[27], and for a conditional knockout (floxed) mutation at the *Hcrtr2* locus (*Hcrtr2*[tm1.1Ava])[54]. All animals were between 12 and 14 weeks old during the time of experiments and animal procedures followed Swiss federal laws and were approved by the State of Vaud Veterinary Office. At all times, care was taken to minimize animal discomfort and avoid pain.

**Mice surgery**. *Hcrtr1/Hcrtr2* double-floxed homozygous mice were stereotaxically injected with *AAV-EF1a-mCherry-IRES-Cre-WPRE* virus (University of North Carolina Vector Core) in the dorsal raphe using a 25 degree angle and the following coordinates (AP, 4.5 mm; ML, 0 mm; DV, 3.2 mm). A 0.8 µl volume of viral particle suspension at a titer of $1.7 \times 10^{12}$ was injected. EEG/EMG electrodes were then implanted during the same surgery as previously described[55,56]. A minimum of two weeks separated surgery and the beginning of recordings to allow recovery and viral particles expression.

**Data acquisition**. Mice after surgery were connected to EMBLA™ hardware for signal acquisition and Somnologica-3™ (Medcare) software for data analysis. Video tracking of cataplexy attacks was based on Vassalli et al.[25].

**Immunofluorescence**. Was carried out according to Li et al.[27]. Antibodies used were Mouse-anti-TPH (sigma, Cat# T0678), Rat-anti-mCHERRY (life technologies, Cat# M11217), Chicken-anti-GFP (Aves Labs, Cat# 1020), and Rabbit-anti-CRE (Novagen, Cat# 69050-3). Coronal brain sections were cut at 20 µm thickness from −4.04 to −4.96 mm relative to Bregma, which includes the anterior and posterior DR regions. To include the complete region of interest, images were scanned in tiling mode set as rectangular grid. All images were acquired on an inverted Zeiss LSM780 confocal laser-scanning microscope (405, 488, and 561 nm lasers) using a ×40 oil lens. ImageJ software was used for image processing and cell counting. Quantification was performed on tile-scanned images generated by confocal microscopy and areas of quantification were chosen according to the Paxinos and Franklin atlas of the mouse brain. For each animal, 3–4 sections collected at −4.04, −4.24, −4.48, and 4.84 mm were analyzed.

**Quantitative PCR**. Quantitative-Real-Time PCR for the analysis of gene expression was carried out by real-time qPCR in an ABI prism HT7900 detection system (in triplicate). Brain samples were collected and preserved at −80 °C until RNA extraction. Total RNA was isolated from hypothalami using the RNeasy Micro kit (QIAGEN 74004). Integrity and quality of RNA samples was verified using a Nanodrop (ND- 1000) spectrophotometer. Reverse transcription was carried out based on Promega reverse transcription Kit by random hexamer or oligo-dT primers according to the manufacturer's instructions. Amplification was performed with SYBER green assay kit.

**Vigilance state analysis**. MATLAB scripts were developed to quantify wakefulness, NREMS, REMS and cataplexy episode numbers, durations and vigilance state fragmentations for which the criteria were described before[57,58]. REMS latency was calculated as the time from the first consolidated bout of NREMS (at least three consecutive epochs), indicating sleep-onset, to the first two consecutive REM epochs during the BL dark periods, or immediately after SD. Cataplexy number and duration analysis were based on our previously reported criteria[25]. TDW analysis was performed according to Vassalli and Franken[26].

**Power spectral analysis**. Using Somnologica-3™ (Medcare) software, each 4 s epoch of the EEG signal was subjected to discrete Fourier transform to determine EEG power density (spectra 0–90 Hz at a frequency resolution of 0.25 Hz). Artifact free, same-state–flanked 4 s epochs were used to calculate the mean EEG spectral profiles for each behavioral states and time intervals. To account for differences among animals in absolute EEG power, the mean values of spectral profiles were represented as percentage of a BL EEG power reference value (100%), calculated for each mouse across BL day one and across all states by summation of the power in 0.75–47.5 Hz frequency bins. To control for differences in the amount of wakefulness, NREMS and REMS, this reference was weighted so that the relative contribution of each state was identical for all mice[56].

**Time-course analysis**. To investigate the dynamic changes in the EEG power in specific frequency bands across day and night, the time-course of the activity of that frequency band was computed for 4 s epochs scored as state of interest (NREMS, REMS and wake). To this end, the number of epochs for each state was divided into percentiles, so that in each percentile approximately the same number of epochs was allocated. The number of percentiles for NREMS and REMS were: 12 for light periods, 6 for dark periods, and 8 for the 6-h light period after SD; for waking state: 6 for light periods, 12 for dark periods, 8 for the 6-h light SD, and 4 for the 6-h light after SD. The EEG power for the frequency range of interest was averaged for each individual percentile and then normalized, depending on the type of analysis, to the BL total power for each state or to power reached at ZT8-12 for delta frequency band in slow-wave sleep. Like power density analysis, single epochs were excluded and only power values of the epochs that themselves, as well as the two adjacent ones, were scored as artifact-free same-state were included in the analysis. MATLAB scripts were developed to analyze the data.

**Statistical analysis**. Animals from all genotypes were randomly distributed for sleep recording sessions. Investigators involved in performing SD and scoring of sleep recordings were blinded as to the animals' genotype. Results are expressed as mean ± SEM or mean ± SD. Statistical analyses were performed using GraphPad prism 8.4.2. and statistical significance of comparisons was determined by t-test, one- or two-way ANOVA and exact *P*, *F*, *t* and *df* values are reported in figure legends. Significant ANOVA analyses were followed by Dunnett's or Tukey multiple comparison post hoc tests. All statistical analyses were reviewed by our institutional statistician.

**Reporting summary**. Further information on research design is available in the Nature Research Reporting Summary linked to this article.

## Data availability

The source data underlying Figs. 1–6 and also Supplementary Figures are provided as a Source Data file. All data are available from the corresponding author upon reasonable request.

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

## Acknowledgements

This work was supported by the Swiss National Science Foundation (grants 173126 to M.T. and 182613 to A.V.).

## Author contributions

Conceptualization: M.T., A.S., and A.V.; investigations: A.S., S.L., and M-L.P.; MATLAB coding and data analysis: A.S.; writing and editing: A.S., A.V., and M.T.

## Competing interests

The authors declare no competing interests.

## Additional information

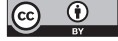

