## [Peer Review File · Nature Communications]

Reviewers' Comments:

Reviewer #1:

Remarks to the Author:

The authors compared the sleep phenotypes of 5HTT KO, Hcrt KO, and double KO mice and found interactions of serotonin and Hcrt systems in the regulation of REM sleep and cataplexy, as well as sleep need and multiple aspects of EEG. Especially, the finding that 5HTT deletion drastically suppressed cataplexy in narcoleptic Hcrt KO mice is interesting and clinically important. While overall these studies are well done, I have several concerns that should be addressed.

First of all, all results are just descriptive and the scientific advances generated by the study are limited. As discussed in the manuscript, dysregulation of REM sleep in Hcrt KO and 5HTT KO mice have been already known. Also, anti-cataplectic effects of SSRIs are known well. Therefore, addition of some mechanistic analyses concerning circuits and molecules underlying the interaction of Hcrt and serotonin systems and/or differential regulation between cataplexy and REM sleep would substantially improve the quality of study.

DR_Hcrt1&2-CKO mice showed a mild phenotype of sleep regulation that was different from other KOs examined in the study. To evaluate the deletion of receptor genes, the authors confirmed the specific and efficient expression of mCHERRY, which was co-expressed with Cre by an AAV vector, and GFP, which was expressed upon deletion of a floxed allele. However, efficient expression of these fluorescent proteins does not necessarily guarantee the efficient deletion of Hcrt1 and Hcrt2. AAV-mCherry-IRES-Cre was used but translation of Cre via IRES should be much less than that of mCherry. In addition, although GFP expression occurred when at least one of 4 alleles, i.e. 2 floxed Hcrt1 and 2 floxed Hcrt2 alleles, was deleted by Cre, deletion of all 4 alleles were required to disrupt HCRT signaling in individual DR serotonin neurons, most of which express both Hcrt1 and 2. Furthermore, disruption of HCRT signaling in almost all cells (of the relevant brain regions) is required for the emergence of cataplexy (Tabuchi et al, J Neurosci 2014). Collectively, due to an incomplete gene deletion, DR_Hcrt1&2-CKO mice in the study might not demonstrate the sleep abnormalities that would appear when HCRT in/out to DR serotonin neurons are truly disrupted.

Reviewer #2:

Remarks to the Author:

This is a good research manuscript on the sleep disorder narcolepsy. By using several genetic animal models and detail sleep data analysis, this study confirmed the critical role of the serotonin in controlling cataplexy, which was initially reported in Hasegawa's studies. Additionally, this study answered other important questions such as the dissociation between REM sleep and cataplexy, which is a hot topic on understanding the brain circuitry for cataplexy. The delta power and spectra analysis in various phenotypes also provide new clues for sleep homeostasis regulation. The followings are some suggestions for the authors. Overall, it is a novel study addressing an important issue in sleep science.

1. HCRT knockout mice show fewer cataplexy bouts after generations due to the behavioral compensatory effect. This is why most studies focusing on cataplexy have to use various stimuli to trigger more cataplexy, especially when anti-cataplexy effects are to be studied. The low cataplexy incidence in this study might cause the full strength of cataplexy suppressing underestimated. Thus, study on triggered cataplexy is strongly recommended to make the conclusion more convincing.

2. A graph of synchronized EEG/EMG with power density to show a typical Cataplexy bout should be included in the figures.

3. Need more information on the statistical analysis of various parameters. Since this is a study with multiple groups, ANOVA alone is not enough.

Reviewer #3:
Remarks to the Author:

Based on results from crossing the 5HTT KO and Hcrt KO mice, the authors would like to make the case that (1) deleting the 5HT transporter in hypocretin knockout mice suppresses cataplexy while dramatically increasing REM sleep; (2) double knock out mice show a significant deficit in the buildup of sleep need; (3) deletion of hypocretin receptors in dorsal raphe 5HT neurons in adult wild type mice does not induce cataplexy but consolidates REM sleep; and (4) cataplexy and REM sleep are regulated by different mechanisms and that both states and sleep need are regulated by the Hcrt input into 5HT neurons. Although the text is rough in places as indicated by my (overly) extensive edits below, the figures are particularly well-prepared. However, I have some problems to be convinced of the authors' conclusions for the following reasons:

- 1) The data presented are based on constitutive single and double knockouts in which we can expect that considerable compensation has occurred during development. This design is well-known to be problematic as the reader is uncertain whether the results obtained are due to the deletion of the gene(s) of interest or to the manner in which the brain has responded to the absence of that gene (or genes).
- 2) The arguments about differential effects on REM sleep are largely based on the data presented in Fig. 1 which need further clarification, as indicated in my Specific Comments below. In particular, the calculation of REM latency (Fig. 1E) is obscure.
- 3) The authors make the argument that elimination of the 5HTT results in stabilization of REM sleep. Although the 5HTT KO has been around for nearly 2 decades, there is little discussion of the previously-described phenotype(s) of this mouse and whether any of the known phenotypes might affect REM Sleep. Also, as stated below, another interpretation of the results presented is that the apparent "REM stabilization" of the 5HTT KO may be a secondary consequence of a defect in the ability of this strain to terminate REM and/or transition between states.
- 4) The cataplexy analyses in Fig. 2A have limited utility as stated below. There are really only two measures of interest: how frequently do cataplexy bouts occur and how long does a bout last? A third item of interest is when these bouts occur during the 24 hr period. The upper panel in Fig. 2A address the frequency issue but the duration issue is not adequately addressed. The "Duration (min/hr)" parameter presented in the lower panel has limited utility to assess the bout duration; what is really needed is an analysis such as that in Fig. 2C.
- 5) WRT the authors' argument that "there is a deficient build-up in homeostatic sleep need during the major wakefulness period": how do the authors account for the fact that the peak EEG delta power is indistinguishable in the DKO from any other strain? To me, this indicates that homeostatic sleep need (τ_{ai}) has accumulated to the same degree but the faster kinetics of the decline of EEG delta power in the DKO suggests that the absence of these genes has affected τ_{ad} , to use the terminology described in the Franken et al. (2001) J Neurosci paper.
- 6) Either there is an error in Fig. 5C or I have completely missed the authors' point.
- 7) The authors could do a better job of integrating or contrasting their results with that of Hasegawa by pointing out the difference in strategies: local restoration of a gene encoding a receptor in a KO vs. local cKO of a receptor in which the receptors elsewhere in the brain should be intact. For example, from a technical point-of-view, how does the degree of receptor expression/deletion affect the results or their interpretation? What about use of different background strains (e.g., reintroduction into a KO strain vs. deletion in a (somewhat) normal background strain)?

There are other points to be addressed in the Specific Comments below.

Specific Comments

I. 17: I don't agree with this statement. As indicated below, an alternative explanation is that DKO mice may actually be more "efficient" in responding to SD, which results in lower EEG DP during the dark phase.

- l. 92: Sentence would be clearer as "In contrast, the increase in REM sleep during the dark phase..."
- l. 95: "the effect" should be "an effect".
- l. 97: "...suggesting further stabilization of REM sleep..." and Fig. 1C. These results could also be viewed as a defect in the termination of REM sleep or in the transition to another state.
- l. 111: "determine" would sound better than "know" here
- l. 113: How you measured "REM latency" is critical and not well explained in l. 400 of Methods.
- l. 120-122: This is an oddly phrased sentence as "Hcrt loss during the dark period" sounds like a conditional deletion that is occurring at one time of day.
- l. 127 and throughout: I presume "BS" means "baseline"; if so, "BL" would be better.
- l. 130-133: "Time spend in cataplexy also is reduced...". Since there are fewer bouts, it is not surprising that the time spent in cataplexy would be reduced when expressed as min/hr. What would be more informative would be the duration of cataplexy events (e.g., min/event), with the mean or median and duration of individual events illustrated rather than simply the mean +/-SEM. This would be a more informative way to determine whether the absence of either the 5HTT or Hcrt affects the "stability" of cataplexy once it has been initiated.
- l. 134: "REM latency": Again, how measured?
- l. 135: Here and throughout text: probably better to use the past tense (e.g., "pronounced").
- l. 140: "these waking episodes had a shorter duration" would probably be better as "these waking episodes were of shorter duration".
- l. 144-45: "...other narcolepsy with cataplexy-related features..." Since it is not clear what you are referring to here, probably best to insert "such as..." to the end of this phrase.
- l. 152-153: I think the authors are referring to the baseline night in this sentence and, if so, should state so. Since the previous sentence refers to the recovery after SD, it is a bit confusing regarding to which period this sentence refers.
- l. 152: "...the buildup...". This phrasing is misleading as a "buildup" (of what?) is an inference; you report EEG Delta Power and should describe that directly. Are the asterisks in the proper place in during baseline in Fig. 2D? It seems that the blue asterisks should be earlier in the night.
- l. 154: "our" should be "the". More importantly, the entire argument about "Delta power attenuation" after SD in this paragraph is undercut by the pre-existing differences that are evident on the baseline night.
- l. 156: "This indicates that there is a deficient build-up in homeostatic sleep need during the major wakefulness period." Another possibility is that the DKO may actually be more "efficient" in responding to SD, which results in lower EEG DP during the dark phase.
- l. 158: "...NREM spectral analysis in baseline light period..." should be "...NREM spectral analysis during the baseline light period..."
- l. 159: "5HTTKO/KO and DKO mice that spend more time..."; this phrase is unclear: are the authors referring to a subset of 5HTTKO/KO and DKO mice that "spend more time...."? The meaning of this sentence is completely different if the word "that" has been improperly inserted in this sentence.
- l. 160: eliminate comma; use past tense for "express" ("expressed").
- l. 164: use past tense for "behave"
- l. 168: Blunting of 0.5-2.5Hz has also been described in nNOS KO mice by Morairty et al. (2013) PNAS.
- l. 173: Insert "the" before "sleep".
- l. 176: "goal oriented" should be "goal-oriented".
- l. 178: "theta dominated wakefulness" should be "theta-dominated wakefulness".
- l. 179: "Driver" misspelled; add "the" after "During"
- l. 184: "dose" should be "does".
- l. 187: "The interesting finding here..." Perhaps you should add "and paradoxical" after "interesting" since you show that the Hcrt-/- mice have lower TDW during baseline.
- l. 191: eliminate comma.
- l. 203: "cataplexy expressing" should be "cataplexy-expressing".
- l. 226-7: eliminate both commas.
- l. 232: "exhibits" should be singular.
- l. 233: "who" should be "which".
- l. 244: "dorsal" misspelled.

- l. 246: eliminate comma but add one after "end".
- l. 247: Add comma after "mice".
- l. 251: "Tph positive" should be "Tph-positive"; neurons misspelled.
- l. 253: "has" should be "had".
- l. 257: eliminate comma
- l. 266: Unless Fig. 5C has the colors reversed, I believe "increased" in this sentence should be "reduced"; Fig. 5C does not support the claim made in the current sentence.
- l. 280: "dampening" misspelled.
- l. 286: eliminate comma
- l. 287: "unites" should be "units"
- l. 293-8: The Tabuchi et al. (2013) Sleep paper is relevant here.
- l. 300: eliminate comma
- l. 301: "consistent" misspelled
- l. 304: "REM sleep is a normal and cataplexy pathological brain states²⁵, ..." – this sentence needs to be re-written and the comma is inappropriate.
- l. 309: "occurrence" misspelled.
- l. 308-10: This sentence needs to be re-written.
- l. 312-314: It is unclear whether the authors are making this argument based on data in the present paper or their previously published work. If based on the current paper, they should refer to the specific figure that supports their results here.
- l. 340: Add comma after "Altogether"
- l. 349: The change in Qfrp levels is not significant and should be deleted from this sentence.
- l. 355: Add comma after "Recently"
- l. 362: Add comma after "summary". Also, since reduction of EEG SWA in the 0.5-2.5Hz range has also been described in nNOS KO mice (Morairty et al., 2013, PNAS), could this range of the EEG be particularly susceptible to disruption?
- l. 372 and elsewhere in text: Probably better to have citations after the ")" in both cases.
- l. 374: Add comma after "times".
- l. 379: Parentheses inappropriate here.
- l. 386: "are" should be "were"
- l. 400: Criteria to calculate REM latency (critical for Fig. 1E) needs further explanation as the epoch size used for EEG scoring is not described. I would not accept a measurement of REM latency that is based on "...the time from the first epoch of NREM sleep occurred after wakefulness to the first REM epoch" as being an accurate measure because a single mis-scored NREM epoch (whether of 4- or 10-sec duration) could result in a misleading calculation of REM latency. To avoid such artifactual errors, I would suggest a rule of 3 consecutively scored epochs of NREM sleep followed by at least two consecutively scored REM epochs as necessary to provide an accurate measure of REM latency.
- l. 413: "is" should be "was".
- l. 419-21: This entire section and this sentence, in particular, needs to be re-written with proper punctuation to provide clarity. Use past rather than present tense throughout this paragraph.
- l. 426: What is the "upper part"?
- l. 433: "achieve" would be better as "calculate" and "accumulated" as "cumulative".
- l. 450: Why use "baseline 24 hours" instead of only the dark phase when cataplexy is more frequent?
- l. 454: eliminate comma
- l. 489: "in" should be "of"
- l. 490: "is" should be "are"
- l. 498: "induces" would be better as "enhances".

Fig. 1: There seems to be a disconnect between the REM sleep values reported in Figs. 1A, 1B and 1D. In Figs 1A and 1B, there is no difference across groups in the amount of REM sleep during the recovery in the light period. However, Fig. 1D clearly shows differences among groups relative to their respective baseline days. Although the authors underscore these differences in the text, the absence of a difference across groups in Figs. 1A and 1B raises the question of the significance of the analyses illustrated in Fig. 1D.

Fig. S1: Add color code to figure. Use a different color to denote the "Sleep deprivation" as the color chosen is too close to that of one of the experimental groups.

Reviewer #1 (Remarks to the Author):

The authors compared the sleep phenotypes of 5HTT KO, Hcrt KO, and double KO mice and found interactions of serotonin and Hcrt systems in the regulation of REM sleep and cataplexy, as well as sleep need and multiple aspects of EEG. Especially, the finding that 5HTT deletion drastically suppressed cataplexy in narcoleptic Hcrt KO mice is interesting and clinically important. While overall these studies are well done, I have several concerns that should be addressed.

We thank our reviewer for pointing out the clinical importance of our findings.

First of all, all results are just descriptive and the scientific advances generated by the study are limited. As discussed in the manuscript, dysregulation of REM sleep in Hcrt KO and 5HTT KO mice have been already known. Also, anti-cataplectic effects of SSRIs are known well. Therefore, addition of some mechanistic analyses concerning circuits and molecules underlying the interaction of Hcrt and serotonin systems and/or differential regulation between cataplexy and REM sleep would substantially improve the quality of study.

We agree that dysregulation of REM sleep has been documented in Hcrt KO and 5HTT KO mice, as well as the anticataplectic effects of SSRIs. Nevertheless, not everybody (especially among clinicians) agrees that the main player in cataplexy is the serotonergic rather than the noradrenergic neurotransmission system. As also discussed in our manuscript, most of the pharmacology evidence comes from the work on canine narcolepsy that strongly suggested a critical role for the noradrenergic system in cataplexy. Also, the role of the 5HT in REM sleep regulation is not well-documented (despite the referenced evidence from 5HTT KO mice). Based on the canine and data in other species, the major player in REM sleep regulation is thought to be the cholinergic system. How Hcrt affects REM sleep and through which pathway is not well-understood (it's accepted that Hcrt stimulates monoaminergic systems which inhibit REM sleep, but the individual contribution of 5HT, NA, DA, and HIS remains unclear). The major finding of our work, summarized in the title, is the interaction between Hcrt and 5HT in regulating both REM sleep and cataplexy. Using our models, for the first time, we show that REM sleep and cataplexy follow different underlying mechanisms. We certainly could get closer to mechanisms and attempt to define relevant circuits/brain areas/cell types by, for instance, assessing state-specific 5HT levels in several 5HT target areas, FOS activity, perform molecular or molecular anatomical differential analyses of vlPAG/LDT/subC/amy/other major DR targets in 5HTT-KO, OX-KO vs DKO mice, or look at the activity/expression of specific candidates in REM sleep or cataplexy-associated brain structures in singles vs double mutants (for instance the MCH circuit enhanced in DKO mice to explain increased REM sleep? or is the NA circuit repressed? As reported in Fig. 6, MCH gene expression tends to be increased, although not significantly due to the low number of mice) etc.

However, we would like to pinpoint that such analyses are beyond the scope of our present paper, which, once our leading hypothesis of the critical impact of brain-wide 5HT tone on cataplexy and REM sleep modulation is verified, aims at performing a detailed differential EEG and cataplexy analysis of the 3 homozygous KO mutants, a heterozygous compound and WT, thus setting the ground for future studies.

Based on available data and our findings, the potential mechanistic interpretation concerning circuits and molecules underlying the interaction between Hcrt and 5HT is that 5HT inhibits Hcrt neurons by acting at 5HT_{1A}, while Hcrt activates 5HT neurons by acting at Hcrt_{1&2}. This is now clearly indicated in our discussion.

DR_Hcrt1&2-CKO mice showed a mild phenotype of sleep regulation that was different

from other KOs examined in the study. To evaluate the deletion of receptor genes, the authors confirmed the specific and efficient expression of mCHERRY, which was co-expressed with Cre by an AAV vector, and GFP, which was expressed upon deletion of a floxed allele. However, efficient expression of these fluorescent proteins does not necessarily guarantee the efficient deletion of Hcrtr1 and Hcrtr2. AAV-mCherry-IRES-Cre was used but translation of Cre via IRES should be much less than that of mCherry. In addition, although GFP expression occurred when at least one of 4 alleles, i.e. 2 floxed Hcrtr1 and 2 floxed Hcrtr2 alleles, was deleted by Cre, deletion of all 4 alleles were required to disrupt HCRT signaling in individual DR serotonin neurons, most of which express both Hcrtr1 and 2. Furthermore, disruption of HCRT signaling in almost all cells (of the relevant brain regions) is required for the emergence of cataplexy (Tabuchi et al, J Neurosci 2014). Collectively, due to an incomplete gene deletion, DR_Hcrtr1&2-CKO mice in the study might not demonstrate the sleep abnormalities that would appear when HCRT input to DR serotonin neurons are truly disrupted.

AAV-mCherry-IRES-Cre was used but translation of Cre via IRES should be much less than that of mCherry. We agree with our reviewer that translation of genes via IRES is substantially less than the gene placed upstream, when used in knock-in models as a single copy. Here, both mCherry and Cre are expressed under a constitutive promoter Efla by an AAV construct (multiple copies) and there's no evidence that in such constructions the IRES-Cre activity is problematic. It is commonly accepted that even trace amounts of Cre are sufficient to mediate comprehensive loxP site recombination. In our experiment we used mCherry to document Cre expression in 5HT neurons, and GFP to document Hcrtr1 and 2 gene deletions since our mouse models are designed to express GFP only when the floxed allele undergoes deletion (Vassalli et al, SciTranslMed 2015; Li et al., Sci Rep 8, 15474, 2018). Nevertheless, following this remark, we have double-stained adjacent sections for mCHERRY and Cre and found a strong Cre staining and a substantial colocalization. This information is now added to the text and reported as Supplementary Fig. 3.

Our reviewer is right that in theory recombination of one of the 4 floxed alleles in our Hcrtr1-R2 double floxed mice could activate some level of GFP expression. But, as mentioned above, because of the reported exquisite efficiency of Cre, there are no reasons to believe a floxed allele could remain intact in a cell in which another allele would have efficiently recombined and turned GFP on.

To transduce the DR of our mice, we followed the same viral injection technique as (Hasegawa et al., 2014). In their work, mice were analyzed 2 weeks after injection, which resulted in rescuing cataplexies of Hcrtr1&2^{KO/KO} mice. This study showed that 2 weeks of AAV expression is enough to observe a reliable phenotype. We used AAV-Ef1a-mCherry-IRES-Cre to shut down the HCRT-DR pathway in WT mice in order to, assess if we can ‘de novo’ induce cataplexy-like events in normal mice, rather than decreasing their incidence in mice that already are narcoleptic. Our AAV, a well-established vector for co-expression of Cre recombinase and a reporter, was originally published in Nature Methods (Fenno et al., 2014) and later many studies used similar constructs (Gunaydin et al., 2014; Nieh et al., 2015; Nomura et al., 2015; Sun et al., 2017; Zhang et al., 2016). We agree with our reviewer regarding the Tabuchi paper indicating that “disruption of HCRT signaling in almost all cells (of the relevant brain regions) is required for the emergence of cataplexy”, and we discuss this point in Discussion, but reaching 100% transduction efficiency using viral vectors is not yet achievable.

Finally, given the possibility of volume transmission of 5HT and extra-synaptic neurotransmission, rather than, or in addition to, 5HT effects at synapses, (as reflected in the fact that it commonly takes 4 or more weeks of SSRI treatment in patients to observe an antidepressant effect), it is possible that a longer duration of Hcrtr receptor inactivation in DR would be needed to see cataplexies appear. This possibility is now added to our discussion.

Reviewer #2 (Remarks to the Author):

This is a good research manuscript on the sleep disorder narcolepsy. By using several genetic animal models and detail sleep data analysis, this study confirmed the critical role of the serotonin in controlling cataplexy, which was initially reported in Hasegawa’s studies. Additionally, this study answered other important questions such as the dissociation between REM sleep and cataplexy, which is a hot topic on understanding the brain circuitry for cataplexy. The delta power and spectra analysis in various phenotypes also provide new clues for sleep homeostasis regulation. The followings are some suggestions for the authors. Overall, it is a novel study addressing an important issue in sleep science.

We thank our reviewer for these encouraging comments.

1. HCRT knockout mice show fewer cataplexy bouts after generations due to the behavioral compensatory effect. This is why most studies focusing on cataplexy have to use various stimuli to trigger more cataplexy, especially when anti-cataplexy effects are to be studied. The low cataplexy incidence in this study might cause the full strength of cataplexy suppressing underestimated. Thus, study on triggered cataplexy is strongly recommended to make the conclusion more convincing.

We partially agree with our reviewer but our aim was to investigate just the occurrence of spontaneous cataplexy. In this optic, we believe a significant change in cataplexy attacks (number and duration) reliably answers our question. As we show in Fig. 2c, double KO mice show near absence of cataplexy (with significantly reduced duration). Furthermore, while behavioral compensation is clinically known in human narcolepsy patients, there is no evidence, to our knowledge, that mice develop a behavioral compensatory response to cataplexy, although such compensation may perhaps arise in older mice.

We would like to make three further remarks:

The number of cataplexy episodes we report here are not significantly less than other studies. Our $Hcrt^{KO/KO}$ and $Hcrt^{KO/KO};5HTT^{+/KO}$ mice displayed 12.4 ± 3.3 and 10.25 ± 1.34 cataplexy episodes, respectively, compared to DKO mice (3.2 ± 0.66). This number of spontaneous cataplexy is consistent with previous studies which reported on average between 10-20 spontaneous cataplexies (Burgess et al., 2013; Oishi et al., 2013) per 12hrs of dark period. Spontaneous cataplexy is absent or rare during the light period. We now report in Fig 2c the number of cataplexy attacks during the dark period.

There is some evidence indicating that spontaneous and triggered cataplexy might involve different mechanisms and circuitries. Accordingly, Oishi et al (J. Neurosci. 2013) reported that reversible inhibition of medial prefrontal cortex substantially reduced cataplexy induced by chocolate but had no effect on spontaneous cataplexy. More recently Sun et al. (Elife 2019) reported two distinct GABAergic neuronal groups in the amygdala involved in spontaneous and predator odor-induced cataplexy.

Nevertheless, Burgess et al. (J. Neurosci. 2013) reported that excitotoxic lesions of amygdala significantly reduced spontaneous as well as wheel running and chocolate-induced cataplexy.

Based on these available data, we remain confident that our observation of a significant suppression of spontaneous cataplexy in our DKO mice is highly relevant. We hope that our reviewer would agree that replicating this finding with triggered cataplexy, which will require few months of experiments, would not make our conclusions more convincing.

2. A graph of synchronized EEG/EMG with power density to show a typical Cataplexy bout should be included in the figures.

We have now included such a graph in Fig. 2a-b.

3. Need more information on the statistical analysis of various parameters. Since this is a study with multiple groups, ANOVA alone is not enough.

We have now added a Statistical section to the Methods and report detailed statistical results in our figure legends. Additionally, we have requested our institutional statistician to review all analyses for approval.

Reviewer #3 (Remarks to the Author):

Based on results from crossing the 5HTT KO and Hcrt KO mice, the authors would like to make the case that (1) deleting the 5HT transporter in hypocretin knockout mice suppresses cataplexy while dramatically increasing REM sleep; (2) double knock out mice show a significant deficit in the buildup of sleep need; (3) deletion of hypocretin receptors in dorsal raphe 5HT neurons in adult wild type mice does not induce cataplexy but consolidates REM sleep; and (4) cataplexy and REM sleep are regulated by different mechanisms and that both states and sleep need are regulated by the Hcrt input into 5HT neurons. Although the text is rough in places as indicated by my (overly) extensive edits below, the figures are particularly well-prepared. However, I have some problems to be convinced of the authors' conclusions for the following reasons:

1) The data presented are based on constitutive single and double knockouts in which we can expect that considerable compensation has occurred during development. This design is well-known to be problematic as the reader is uncertain whether the results obtained are due to the deletion of the gene(s) of interest or to the manner in which the brain has responded to the absence of that gene (or genes).

We can only agree with this important and well-known limitation of constitutive KO models. Nevertheless, we believe that comparing our KO mice to appropriate controls and additionally also including heterozygous mice, provide strong evidence for our findings. Note also that making time-specific (at adulthood) conditional mice for 2 or more genes at the same time is highly complex, if possible.

2) The arguments about differential effects on REM sleep are largely based on the data presented in Fig. 1 which need further clarification, as indicated in my Specific Comments below. In particular, the calculation of REM latency (Fig. 1E) is obscure.

See below our new definition of REM latency. Based on our reviewer suggestion, we have recalculated REMS latencies in all genotypes and performed new analysis reported in Fig. 1d.

3) The authors make the argument that elimination of the 5HTT results in stabilization of REM sleep. Although the 5HTT KO has been around for nearly 2 decades, there is little discussion of the previously-described phenotype(s) of this mouse and whether any of the known phenotypes might affect REM Sleep. Also, as stated below, another interpretation of the results presented is that the apparent “REM stabilization” of the 5HTT KO may be a secondary consequence of a defect in the ability of this strain to terminate REM and/or transition between states.

We have now added more information on previous findings on REM sleep in 5HTT KO mice. Note that the mechanisms are largely unknown but believed to be developmentally regulated:

“5HTT knockout mice were reported to have increased basal extracellular serotonin(Mathews et al., 2004), which results in increased REMS (our data and others)(Wisor et al., 2003). Disruption of 5HTT function leads to major pleiotropic central and peripheral phenotypes(Murphy and Lesch, 2008), but whether any of these has a direct impact on REMS quality and quantity is unknown. 5HTT^{KO/KO} mice display altered expression and function of 5-HT1A and 1B receptors(Fabre et al., 2000) and fail to show a delayed increase in REMS under restraint stress, as do WT mice(Rachalski et al., 2009). The latter effect was suggested to result from enhanced hypocretinergic neurotransmission(Rachalski et al., 2009). Our findings however challenge this interpretation, as we found increased REMS in 5HTT^{KO/KO} mice both in presence or absence of functional HCRT neurotransmission. It was proposed that the overexpression of REMS in adult 5HTT^{KO/KO} mice results from excess extracellular 5HT brain levels during development, as early life blockade of 5HT synthesis, or reducing excessive 5HT1AR stimulation by endogenous 5HT, result respectively, in suppression, or limitation of REMS increase(Alexandre et al., 2006). In addition, during development, 5HTT is transiently expressed in the prefrontal cortex(Frazer et al., 2015; Soiza-Reilly et al., 2018) where 5HTT-expressing neurons (which are not serotonergic) coordinate normal prefrontal cortex-DR innervation. Since the prefrontal cortex is the site through which positive emotions trigger cataplexy(Oishi et al., 2013; Vassalli et al., 2013), it can be speculated that developmentally-modified prefrontal cortex by excess 5HT cannot trigger cataplexy.

The suggestion of our reviewer that instead of “stabilization” there must be “a defect in the ability of this line to terminate REM and/or transition between states” is an interesting speculation that we have been happy to follow up by adding the following:

“To test whether REMS stabilization results from an inability of 5HTT^{KO/KO} and DKO mice to terminate REMS bouts, or to switch states, we analyzed transitions into and out of REMS. This analysis (Supplementary Fig. 1c) indicated no difference in transition numbers between 5HTT^{KO/KO} and WT mice, while DKO mice showed significantly more transitions both in and

out of REMS than WT mice. Therefore, the increase in the amount of REMS, as well as in REMS bout length in these mutant mice does not seem to be due to a defect in terminating REMS.”

4) The cataplexy analyses in Fig. 2A have limited utility as stated below. There are really only two measures of interest: how frequently do cataplexy bouts occur and how long does a bout last? A third item of interest is when these bouts occur during the 24 hr period. The upper panel in Fig. 2A address the frequency issue but the duration issue is not adequately addressed. The “Duration (min/hr)” parameter presented in the lower panel has limited utility to assess the bout duration; what is really needed is an analysis such as that in Fig. 2C.

We have now analyzed the number of cataplexy episodes based on their bout length (Fig. 2c) and the results clearly show a dramatic decrease in bout length for the most frequent episodes.

5) WRT the authors' argument that "there is a deficient build-up in homeostatic sleep need during the major wakefulness period": how do the authors account for the fact that the peak EEG delta power is indistinguishable in the DKO from any other strain? To me, this indicates that homeostatic sleep need (τ_{ui}) has accumulated to the same degree but the faster kinetics of the decline of EEG delta power in the DKO suggests that the absence of these genes has affected τ_{ud} , to use the terminology described in the Franken et al. (2001) J Neurosci paper. *We thank our reviewer for this critical observation. It is correct as shown in Fig. 2D (now 2f) that DKO mice reach the same level of delta power at the end of the baseline dark period. Our interpretation of a defect in delta build-up is based on the fact that all other strains rapidly build up delta power during the first part of the dark period (spent mostly in wakefulness) and then dissipate it during the second part of the dark period (mostly in NREM sleep), while DKO mice (although with the same amount of sleep and wakefulness), increase delta power very slowly and without dissipation. Accordingly, they reach the same level as other strains. As we have shown in 2001 (Franken et al., J. Neurosci), τ_{ui} and τ_{ud} are function of sleep and wakefulness distribution at each time point and in mice with short periods of sleep and wakefulness, these constants are varying every hour (or even less). Here, we did not perform any simulation but our Fig. 2D (now 2f) clearly indicates that all strains except DKO have a much larger τ_{ui} during the first part of the dark period (they build up sleep need normally). We also observed a slower build-up of sleep need in Hcrt KO mice (Fig 2D, now 2f) as we had previously reported (Vassalli and Franken, PNAS 2018) but this is by far less pronounced as compared to DKO mice. We hope this clarifies this important point.*

6) Either there is an error in Fig. 5C or I have completely missed the authors' point. *We are sorry for this confusion. In Fig. 5C there is a significant increase in the low delta frequency (0.75Hz) during both dark and light periods, which becomes more apparent after SD. There is also a significant decrease between 2 and 7.75Hz during the light and between 1.75 and 8.5Hz during the dark period. We have now highlighted the increase in the low delta frequency for better visibility. This figure is now modified for clarification.*

7) The authors could do a better job of integrating or contrasting their results with that of Hasegawa by pointing out the difference in strategies: local restoration of a gene encoding a receptor in a KO vs. local cKO of a receptor in which the receptors elsewhere in the brain should be intact. For example, from a technical point-of-view, how does the degree of receptor expression/deletion affect the results or their interpretation? What about use of different background strains (e.g., reintroduction into a KO strain vs. deletion in a (somewhat) normal background strain)?

We believe our results are different but complementary to the findings of Hasegawa et al.. In Hasegawa work the mice are constitutively receptor deficient everywhere in the brain and they restored them in DR. In our work, wild-type mice are acutely deleted for these receptors only in DR making any direct comparison difficult. Since Hasegawa findings indicates that restoring Hcrt receptors in DR while the rest is nonfunctional, our aim was to see if deleting these receptors only in DR may provoke cataplexy. Regarding the “degree of receptor expression/deletion” we believe this might be very similar between Hasegawa and our work because we have used nearly the exact technique (DR AAV injection).

Reintroduction into a KO strain is what has been done by Hasegawa and deletion by us! Finally, one cannot exclude that deleting these receptors in DR may produce cataplexy after an extend time (we have now added this possibility to our discussion, see also our comment at the end of our answers to Reviewer#1). As we indicate in our paper: acute deletion does not produce cataplexy.

There are other points to be addressed in the Specific Comments below.

Specific Comments

l. 17: I don't agree with this statement. As indicated below, an alternative explanation is that DKO mice may actually be more “efficient” in responding to SD, which results in lower EEG DP during the dark phase.

Please see our response to Point 5. The response to SD is intact in all strains studied here. The increase in delta power induced by SD can be normal while its spontaneous build-up may be compromised. We have previously reported this finding in Hcrt KO mice (Vassalli and Franken, PNAS 2017).

l. 92: Sentence would be clearer as “In contrast, the increase in REM sleep during the dark phase...”

Changed as suggested.

l. 95: “the effect” should be “an effect”.

Changed to: “...a finding in accordance with previous reports”

l. 97: “...suggesting further stabilization of REM sleep...” and Fig. 1C. These results could also be viewed as a defect in the termination of REM sleep or in the transition to another state.

See our answer to Point 3.

l. 111: “determine” would sound better than “know” here

Changed as suggested.

l. 113: How you measured “REM latency” is critical and not well explained in l. 400 of Methods.

REM latency is now better explained:

“REM latency was calculated as the time from the first consolidated bout of NREM sleep (3 consecutive epochs), indicating sleep onset, to the first two consecutive REM epochs.” As suggested by our reviewer.

l. 120-122: This is an oddly phrased sentence as “Hcrt loss during the dark period” sounds like a conditional deletion that is occurring at one time of day.

The sentence was changed into

“Altogether, these data indicate that 5HTT loss during the light period, whether in the presence (5HTT^{KO/KO}) or absence (DKO) of HCRT signaling, and HCRT loss during the dark period, whether in normal (Hcrt^{KO/KO}) or altered 5HT signaling (5HTT^{+ /KO}; Hcrt^{KO/KO} and DKO), lead to an increase in REMS propensity, suggesting an important role of the 5HT and HCRT systems in REMS regulation.”

l. 127 and throughout: I presume “BS” means “baseline”; if so, “BL” would be better

BS is changed to BL throughout.

l. 130-133: “Time spend in cataplexy also is reduced...”. Since there are fewer bouts, it is not surprising that the time spent in cataplexy would be reduced when expressed as min/hr. What would be more informative would be the duration of cataplexy events (e.g., min/event), with the mean or median and duration of individual events illustrated rather than simply the mean +/-SEM. This would be a more informative way to determine whether the absence of either the 5HTT or Hcrt affects the “stability” of cataplexy once it has been initiated.

Bout length analysis is now added as Fig. 2c (see above, under remark 4).

l. 134: “REM latency”: Again, how measured?

See above, l.113.

l. 135: Here and throughout text: probably better to use the past tense (e.g., “pronounced”).

Changed as suggested.

l. 140: “these waking episodes had a shorter duration” would probably be better as “these waking episodes were of shorter duration”.

Changed as suggested.

l. 144-45: "...other narcolepsy with cataplexy-related features..." Since it is not clear what you are referring to here, probably best to insert "such as..." to the end of this phrase.

Changed as suggested.

l. 152-153: I think the authors are referring to the baseline night in this sentence and, if so, should state so. Since the previous sentence refers to the recovery after SD, it is a bit confusing regarding to which period this sentence refers.

The sentence was changed for clarification (early dark period of the baseline).

l. 152: "...the buildup...". This phrasing is misleading as a "buildup" (of what?) is an inference; you report EEG Delta Power and should describe that directly. Are the asterisks in the proper place in during baseline in Fig. 2D? It seems that the blue asterisks should be earlier in the night.

The sentence was changed to: However, the build-up in delta power...

The figure was changed to clearly indicate the earlier significant difference. Note that the first percentile is significant at $p < 0.001$, therefore indicated with 3 asterisks of a different color, as explained in the legend.

l. 154: "our" should be "the". More importantly, the entire argument about "Delta power attenuation" after SD in this paragraph is undercut by the pre-existing differences that are evident on the baseline night.

Changed as suggested:

"However, the build-up during early baseline dark period was greatly attenuated in all genotypes compared to WT mice, especially in DKO mice (Figure 2f)."

l. 156: "This indicates that there is a deficient build-up in homeostatic sleep need during the major wakefulness period." Another possibility is that the DKO may actually be more "efficient" in responding to SD, which results in lower EEG DP during the dark phase.

See our answer above regarding the difference between the response to SD and spontaneous sleep need build-up.

l. 158: "...NREM spectral analysis in baseline light period..." should be "...NREM spectral analysis during the baseline light period..."

Changed as suggested.

l. 159: “5HTTKO/KO and DKO mice that spend more time...”; this phrase is unclear: are the authors referring to a subset of 5HTTKO/KO and DKO mice that “spend more time...”? The meaning of this sentence is completely different if the word “that” has been improperly inserted in this sentence.

This sentence was changed to: 5HTT^{KO/KO} and DKO mice spent more time in REMS during the light period and expressed higher slow-delta (1-2Hz) activity relative to WT, while fast-delta (3-4Hz) frequencies were mainly increased in 5HTT^{+ /KO};Hcrt^{KO/KO}, Hcrt^{KO/KO} and DKO mice (Fig. 2g, left).

l. 160: eliminate comma; use past tense for “express” (“expressed”).

Changed as above.

l. 164: use past tense for “behave”

Changed as above.

l. 168: Blunting of 0.5-2.5Hz has also be described in nNOS KO mice by Morairty et al. (2013) PNAS.

Reference to Morairty et al is now added in the same sentence.

l. 173: Insert “the” before “sleep”.

Changed as suggested.

l. 176: “goal oriented” should be “goal-oriented”.

Changed as suggested.

l. 178: “theta dominated wakefulness” should be “theta-dominated wakefulness”.

Changed as suggested.

l. 179: “Driver” misspelled; add “the” after “During”

Changed as suggested.

l. 184: “dose” should be “does”.

Changed as suggested.

l. 187: “The interesting finding here...” Perhaps you should add “and paradoxical” after “interesting” since you show that the Hcrt-/- mice have lower TDW during baseline.

Changed as suggested.

l. 191: eliminate comma.

Changed as suggested.

l. 203: “cataplexy expressing” should be “cataplexy-expressing”.

Changed as suggested.

l. 226-7: eliminate both commas.

Changed as suggested.

l. 232: “exhibits” should be singular.

Changed as suggested.

1. 233: “who” should be “which”.

Changed as suggested.

1. 244: “dorsal” misspelled.

Changed to DR.

1. 246: eliminate comma but add one after “end”.

Changed as suggested.

1. 247: Add comma after “mice”.

Changed as suggested.

1. 251: “Tph positive” should be “Tph-positive”; neurons misspelled.

Changed as suggested.

1. 253: “has” should be “had”.

Changed as suggested.

1. 257: eliminate comma

Changed as suggested.

1. 266: Unless Fig. 5C has the colors reversed, I believe “increased” in this sentence should be “reduced”; Fig. 5C does not support the claim made in the current sentence.

See our response to Point 6.

1. 280: “dampening” misspelled.

Changed as suggested.

1. 286: eliminate comma

Changed as suggested.

1. 287: “unites” should be “units”

Changed as suggested.

1. 293-8: The Tabuchi et al. (2013) Sleep paper is relevant here.

Reference added.

1. 300: eliminate comma

Changed as suggested.

1. 301: “consistent” misspelled

Changed as suggested.

1. 304: “REM sleep is a normal and cataplexy pathological brain states²⁵, ...” – this sentence needs to be re-written and the comma is inappropriate.

The sentence was changed to “REMS and cataplexy are different brain states (Vassalli et al., 2013) that share many similarities including muscle atonia and high EEG theta activity”

l. 309: “occurrence” misspelled.

Changed above.

l. 308-10: This sentence needs to be re-written.

The sentence was changed to: DKO mice exhibit largely reduced cataplexy bouts during the dark period (when nearly all cataplexy episodes occur), while they spend significantly more time in REM sleep.

l. 312-314: It is unclear whether the authors are making this argument based on data in the present paper or their previously published work. If based on the current paper, they should refer to the specific figure that supports their results here.

The sentence was changed to: “Note also that we previously reported that high-amplitude theta-frequency paroxysmal events occurring during cataplexy of mice and children are, unlike REM-sleep theta, confined to the prefrontal cortex in mice (Vassalli et al., 2013).”

l. 340: Add comma after “Altogether”

Changed as suggested.

l. 349: The change in Qfrp levels is not significant and should be deleted from this sentence.

Changed as suggested.

l. 355: Add comma after “Recently”

Changed as suggested.

l. 362: Add comma after “summary”. Also, since reduction of EEG SWA in the 0.5-2.5Hz range has also been described in nNOS KO mice (Morairty et al., 2013, PNAS), could this range of the EEG be particularly susceptible to disruption?

Changed as suggested.

Recent evidence suggests that the slow (< 2 Hz) and fast (2-4Hz) EEG SWA are differentially regulated (Morairty et al., PNAS 2017; Vassalli and Franken, PNAS 2017, Hubbard et al. <https://www.biorxiv.org/content/10.1101/748871v2>, and present work). The slow delta activity is affected by sleep deprivation while the fast delta seems unrelated to time-spent-awake. These findings suggest that changes in the slow EEG activity constitute the most relevant proxy for the sleep homeostatic process. A sentence is now added and Morairty et al is referenced.

l. 372 and elsewhere in text: Probably better to have citations after the “)” in both cases.

Changed as suggested.

l. 374: Add comma after “times”.

Changed as suggested.

l. 379: Parentheses inappropriate here.

Changed as suggested.

l. 386: “are” should be “were”

Changed as suggested.

l. 400: Criteria to calculate REM latency (critical for Fig. 1E) needs further explanation as the epoch size used for EEG scoring is not described. I would not accept a measurement of REM latency that is based on "...the time from the first epoch of NREM sleep occurred after wakefulness to the first REM epoch" as being an accurate measure because a single mis-scored NREM epoch (whether of 4- or 10-sec duration) could result in a misleading calculation of REM latency. To avoid such artifactual errors, I would suggest a rule of 3 consecutively scored epochs of NREM sleep followed by at least two consecutively scored REM epochs as necessary to provide an accurate measure of REM latency.

Changed as suggested (also see above l.113):

"REMS latency was calculated as the time from the first consolidated bout of NREMS (at least 3 consecutive epochs), indicating sleep onset, to the first two consecutive REM epochs during the baseline dark periods, or immediately after SD."

l. 413: "is" should be "was".

Changed as suggested.

l. 419-21: This entire section and this sentence, in particular, needs to be re-written with proper punctuation to provide clarity. Use past rather than present tense throughout this paragraph.

This section was re-written.

l. 426: What is the "upper part"?

"Reference calculated in upper part" is changed to "baseline total power"

l. 433: "achieve" would be better as "calculate" and "accumulated" as "cumulative".

Changed as suggested.

l. 450: Why use "baseline 24 hours" instead of only the dark phase when cataplexy is more frequent?

As indicated above, we now report the 12h dark period for cataplexy. The sentence was changed to "are presented for the 12h dark period."

l. 454: eliminate comma

Changed as suggested.

l. 489: "in" should be "of"

Changed as suggested.

l. 490: "is" should be "are"

Changed as suggested.

l. 498: "induces" would be better as "enhances".

Changed as suggested.

Fig. 1: There seems to be a disconnect between the REM sleep values reported in Figs. 1A, 1B and 1D. In Figs 1A and 1B, there is no difference across groups in the amount of REM

sleep during the recovery in the light period. However, Fig. 1D clearly shows differences among groups relative to their respective baseline days. Although the authors underscore these differences in the text, the absence of a difference across groups in Figs. 1A and 1B raises the question of the significance of the analyses illustrated in Fig. 1D.

Fig. 1 is now changed and 1D is removed.

Fig. S1: Add color code to figure. Use a different color to denote the “Sleep deprivation” as the color chosen is too close to that of one of the experimental groups.

Changed as suggested.

Alexandre, C., Popa, D., Fabre, V., Bouali, S., Venault, P., Lesch, K.P., Hamon, M., and Adrien, J. (2006). Early life blockade of 5-hydroxytryptamine 1A receptors normalizes sleep and depression-like behavior in adult knock-out mice lacking the serotonin transporter. *J Neurosci* 26, 5554-5564.

Burgess, C.R., Oishi, Y., Mochizuki, T., Peever, J.H., and Scammell, T.E. (2013). Amygdala lesions reduce cataplexy in orexin knock-out mice. *J Neurosci* 33, 9734-9742.

Fabre, V., Beaufour, C., Evrard, A., Rioux, A., Hanoun, N., Lesch, K.P., Murphy, D.L., Lanfumey, L., Hamon, M., and Martres, M.P. (2000). Altered expression and functions of serotonin 5-HT1A and 5-HT1B receptors in knock-out mice lacking the 5-HT transporter. *Eur J Neurosci* 12, 2299-2310.

Fenko, L.E., Mattis, J., Ramakrishnan, C., Hyun, M., Lee, S.Y., He, M., Tucciarone, J., Selimbeyoglu, A., Berndt, A., Grosenick, L., *et al.* (2014). Targeting cells with single vectors using multiple-feature Boolean logic. *Nat Methods* 11, 763-772.

Frazer, S., Otomo, K., and Dayer, A. (2015). Early-life serotonin dysregulation affects the migration and positioning of cortical interneuron subtypes. *Transl Psychiatry* 5, e644.

Gunaydin, L.A., Grosenick, L., Finkelstein, J.C., Kauvar, I.V., Fenko, L.E., Adhikari, A., Lammel, S., Mirzabekov, J.J., Airan, R.D., Zalocusky, K.A., *et al.* (2014). Natural neural projection dynamics underlying social behavior. *Cell* 157, 1535-1551.

Hasegawa, E., Yanagisawa, M., Sakurai, T., and Mieda, M. (2014). Orexin neurons suppress narcolepsy via 2 distinct efferent pathways. *J Clin Invest* 124, 604-616.

Mathews, T.A., Fedele, D.E., Coppelli, F.M., Avila, A.M., Murphy, D.L., and Andrews, A.M. (2004). Gene dose-dependent alterations in extraneuronal serotonin but not dopamine in mice with reduced serotonin transporter expression. *J Neurosci Methods* 140, 169-181.

Murphy, D.L., and Lesch, K.P. (2008). Targeting the murine serotonin transporter: insights into human neurobiology. *Nat Rev Neurosci* 9, 85-96.

Nieh, E.H., Matthews, G.A., Allsop, S.A., Presbrey, K.N., Leppla, C.A., Wichmann, R., Neve, R., Wildes, C.P., and Tye, K.M. (2015). Decoding neural circuits that control compulsive sucrose seeking. *Cell* 160, 528-541.

Nomura, H., Hara, K., Abe, R., Hitora-Imamura, N., Nakayama, R., Sasaki, T., Matsuki, N., and Ikegaya, Y. (2015). Memory formation and retrieval of neuronal silencing in the auditory cortex. *Proc Natl Acad Sci U S A* 112, 9740-9744.

Oishi, Y., Williams, R.H., Agostinelli, L., Arrigoni, E., Fuller, P.M., Mochizuki, T., Saper, C.B., and Scammell, T.E. (2013). Role of the medial prefrontal cortex in cataplexy. *J Neurosci* 33, 9743-9751.

Rachalski, A., Alexandre, C., Bernard, J.F., Saurini, F., Lesch, K.P., Hamon, M., Adrien, J., and Fabre, V. (2009). Altered sleep homeostasis after restraint stress in 5-HTT knock-out male mice: a role for hypocretins. *J Neurosci* 29, 15575-15585.

Soiza-Reilly, M., Meye, F.J., Olusakin, J., Telley, L., Petit, E., Chen, X., Mameli, M., Jabaudon, D., Sze, J.Y., and Gaspar, P. (2018). SSRIs target prefrontal to raphe circuits during development modulating synaptic connectivity and emotional behavior. *Mol Psychiatry*.

Sun, Y., Grieco, S.F., Holmes, T.C., and Xu, X. (2017). Local and Long-Range Circuit Connections to Hilar Mossy Cells in the Dentate Gyrus. *eNeuro* 4.

Vassalli, A., Dellepiane, J.M., Emmenegger, Y., Jimenez, S., Vandi, S., Plazzi, G., Franken, P., and Tafti, M. (2013). Electroencephalogram paroxysmal theta characterizes cataplexy in mice and children. *Brain* 136, 1592-1608.

Wisor, J.P., Wurts, S.W., Hall, F.S., Lesch, K.P., Murphy, D.L., Uhl, G.R., and Edgar, D.M. (2003). Altered rapid eye movement sleep timing in serotonin transporter knockout mice. *Neuroreport* 14, 233-238.

Zhang, S., Xu, M., Chang, W.C., Ma, C., Hoang Do, J.P., Jeong, D., Lei, T., Fan, J.L., and Dan, Y. (2016). Organization of long-range inputs and outputs of frontal cortex for top-down control. *Nat Neurosci* 19, 1733-1742.

Reviewers' Comments:

Reviewer #1:

Remarks to the Author:

The author's revisions partially address the issues raised in the last reviews, but I still have several major concerns that are not yet adequately addressed.

As for DR_Hcrt1&2-CKO mice, I strongly agree with the author's point that reaching 100% transduction efficiency using viral vectors is not yet achievable. Even with Cre driver mice, it is very rare to achieve nearly 100 % excision of floxed alleles. The authors say "It is commonly accepted that even trace amounts of Cre are sufficient to mediate comprehensive loxP site recombination", "there are no reasons to believe a floxed allele could remain intact in a cell in which another allele would have efficiently recombined and turned GFP on", please show the evidence or reference. Cre-induced excisions are probabilistic events, depending on Cre expression level and efficiency of excision of floxed alleles, which greatly varies among different floxed alleles. The partial excision often happens. I feel there is no reason to believe all of 4 alleles of Hcrt1&2 are excised in all mCherry-positive cells. So researchers are often suffered from "partial" deletion. When you see some phenotype by a partial deletion, such as consolidation of REM sleep in DR_Hcrt1&2-CKO mice described in the current study, you can claim that the reduced expression of deleted genes is responsible to the phenotype. But when you don't see a particular phenotype, such as cataplexy, you can not exclude the possibility that remained an expression of the gene is sufficient to prevent the phenotype.

The authors estimated the efficiency of Hcrt1&2 deletion in the DR to be 77.85 % by counting mCherry expression driven by the Cre-expressing AAV vector. So remained expression of Hcrt1&2 in 22.15 % of DR serotonergic neurons may be sufficient to prevent cataplexy, considering observations by Tabuchi et al. I guess that mCherry-positive neurons were counted only in the small area of one section indicated by the box in Fig. 4b, because I see many TPH-positive;mCherry-negative (green) neurons in the dorsal area of DR closed to the aqueduct. However, the efficiency of deletion should be estimated for the entire DR. Besides, Hcrt1&2 and Cre-mCherry were expressed not only serotonergic neurons but also in non-serotonergic neurons in the DR, as observed in Fig. 4, which are likely to include local GABAergic cells that may inhibit serotonergic cells. In addition to the efficiency of deletion within DR serotonergic neurons, the specificity of deletion to serotonergic neurons should be estimated.

The authors nicely added double-stained sections for mCherry and Cre in Fig. S3. However, I see many mCherry-positive; Cre-negative (red) cells, which are inconsistent with the authors' claim that all mCherry-positive cells also express Cre. More surprisingly, I see many mCherry-negative; Cre-positive (green) cells, which may raise a question about the specificity of Cre immunostaining.

In Fig. 2d, what are the units of Y axes? Should they be min/h and bouts/h? Are those hourly averages of the entire dark phase? If the units on Y axes are correct, WT spent in wakefulness for about 5 min x 8 times = 40 min on average in the entire dark phase, which sounds odd.

In Fig. 2h, 2i, and 3, why are the measures close between Hcrt KO/KO and DKO but different in 5HTT +/KO; Hcrt KO/KO? Generally speaking, the phenotype of heterozygous mice should be in the range between those of WT and homozygous mice.

Besides, as discussed in the manuscript (line 210-214), the observation is paradoxical.

In lines 265-268, the authors discussed that mechanisms underlying REMS latency, REMS duration, and REMS theta power are differentially regulated. But this just means that there is no clear and explainable correlation between genotypes and sleep phenotypes. It is very difficult to take an overall message from this study.

Lines 345-347, In contrast to cataplexy suppression, a large increase in REMS was found in DKO mice, indicating a dissociation between REMS and cataplexy and suggesting that the two states are regulated by different mechanisms: Such differential regulation of REMS and cataplexy has been already suggested in Hasegawa et al., 2014.

Reviewer #2:

None

Reviewer #3:

Remarks to the Author:

The authors have carefully and appropriately responded to my previous comments.

Reviewer #1 (Remarks to the Author):

The author's revisions partially address the issues raised in the last reviews, but I still have several major concerns that are not yet adequately addressed.

As for DR_Hcrtr1&2-CKO mice, I strongly agree with the author's point that reaching 100% transduction efficiency using viral vectors is not yet achievable. Even with Cre driver mice, it is very rare to achieve nearly 100 % excision of floxed alleles. The authors say "It is commonly accepted that even trace amounts of Cre are sufficient to mediate comprehensive loxP site recombination", "there are no reasons to believe a floxed allele could remain intact in a cell in which another allele would have efficiently recombined and turned GFP on", please show the evidence or reference. Cre-induced excisions are probabilistic events, depending on Cre expression level and efficiency of excision of floxed alleles, which greatly varies among different floxed alleles. The partial excision often happens. I feel there is no reason to believe all of 4 alleles of Hcrtr1&2 are excised in all mCherry-positive cells. So researchers are often suffered from "partial" deletion. When you see some phenotype by a partial deletion, such as consolidation of REM sleep in DR_Hcrtr1&2-cKO mice described in the current study, you can claim that the reduced expression of deleted genes is responsible to the phenotype. But when you don't see a particular phenotype, such as cataplexy, you cannot exclude the possibility that remained an expression of the gene is sufficient to prevent the phenotype.

We thank our reviewer for his/her important comment to improve our manuscript. We can only agree with our reviewer that the recombination is a probabilistic event and since both possibilities (recombined or non-recombined allele) exist, we have now changed the following sentence in our discussion:

"Our result however indicated that acute disruption of the $LH^{HCRT}-DR^{5HT}$ circuit is not sufficient to induce cataplexy in *WT* adult mice, and argues that other HCRT targets can mediate cataplexy protection."

to:

"Our result however indicated that acute disruption of the $LH^{HCRT}-DR^{5HT}$ circuit is not sufficient to induce cataplexy in adult mice. This might be due to the fact that complete disruption of the $LH^{HCRT}-DR^{5HT}$ circuit may not have been reached in our experiment. Not all DR^{5HT} cells were infected and in infected cells not all 4 floxed alleles may have been deleted. Additionally, some cells that showed Cre expression were not TPH positive and these cells may also play a role in cataplexy (for instance local inhibitory GABA cells). A higher level of Cre recombination and the use of cell type-specific promoters are necessary in future experiments."

Please note that we do not claim anything more than what our data show: acute disruption of Hcrtr1&2 in 77.8% of DR neurons in adult wild-type mice does not induce cataplexy.

The authors estimated the efficiency of Hcrtr1&2 deletion in the DR to be 77.85 % by counting mCherry expression driven by the Cre-expressing AAV vector. So remained expression of Hcrtrs in 22.15 % of DR serotonergic neurons may be sufficient to prevent cataplexy, considering observations by Tabuchi et al. I guess that mCherry-positive neurons

were counted only in the small area of one section indicated by the box in Fig. 4b, because I see many TPH-positive;mCherry-negative (green) neurons in the dorsal area of DR closed to the aqueduct. However, the efficiency of deletion should be estimated for the entire DR.

Our estimation of the efficiency of Hcrtr1&2 deletion in the DR as 77.85% (%TPH+ cells that are mCherry-positive) was not based on counting only the indicated area of a single section, but on counting the whole section in 3-4 coronal sections per mouse for a total of 4 mice).

Our text clearly indicates this:

“co-localization of the 5HT cell-specific marker Tryptophan Hydroxylase (TPH) with mCHERRY (77.85% ± 2.95 %, mean ± SD, n=4; Fig. 4b).” (detailed below which was added to the material and methods part).

Coronal brain sections were cut at 20 µm thickness from -4.04 mm to -4.96 mm relative to bregma, which includes the anterior and posterior DR regions. TPH and mCherry immunostaining was performed. To include the complete region of interest, images were scanned in tiling mode set as rectangular grid. Quantification was performed on tile-scanned images generated by confocal microscopy. and areas of quantification were chosen according to the atlas of the mouse brain (Paxinos and Franklin, 2000). For each animal, 3-4 sections collected at -4.04, -4.24, -4.48, and 4.84 mm were analyzed.

Besides, Hcrts and Cre-mCherry were expressed not only serotonergic neurons but also in non-serotonergic neurons in the DR, as observed in Fig. 4, which are likely to include local GABAergic cells that may inhibit serotonergic cells. In addition to the efficiency of deletion within DR serotonergic neurons, the specificity of deletion to serotonergic neurons should be estimated.

This point of our Reviewer is well taken. We now have added new data assessing the specificity of Hcrtr1&2 deletion in DR-5HT neurons. We found that 81.64 ± 5.21 % (mean ± SD, n = 4) of mCherry-positive cells were TPH-positive.

This quantification was performed as the one described above, with 4 mice and 3-4 sections per mouse.

Thus altogether, our estimation for the penetrance (efficiency) of the deletion in DR-5HT cells is 77.86% (fraction of TPH-positive cells that are mCherry-positive) and our estimation of the specificity is 81.64 % (fraction of mCherry positive cells that are TPH-positive). We modified the main text accordingly.

The authors nicely added double-stained sections for mCherry and Cre in Fig. S3. However, I see many mCherry-positive; Cre-negative (red) cells, which are inconsistent with the authors' claim that all mCherry-positive cells also express Cre. More surprisingly, I see many mCherry-negative; Cre-positive (green) cells, which may raise a question about the specificity of Cre immunostaining.

We did not claim that “all mCherry-positive cells also express Cre” but “a substantial colocalization” in our response to reviewers and “We verified that mCherry-immunoreactivity largely reflected Cre recombinase expression” in our main text. Fig. S3 is from a mouse with the following cell counts.

Mouse ID 8172						
Slide	Bregma (mm)	Total CRE	colocalization	Total mCherry	% colocalization/Total CRE	% colocalization/Total mCherry
s02_0103_12x12tile	*-4.24	65	53	67	81.53846154	79.10447761
s02_0104_12x12tile	*-4.36	73	60	86	82.19178082	69.76744186
s02_0202_12x12tile	*-4.84	104	74	100	71.15384615	74
s02_0203_12x12tile	*-4.96	85	66	83	77.64705882	79.51807229
sum		327	253	336	77.37003058	75.29761905

Since the last revision, we performed additional cell counting in 3 mice, which indicated that $73.82 \pm 1.64\%$ (mean \pm SD, $n = 3$) mCherry positive neurons were CRE-positive. Conversely, $77.60 \pm 3.62\%$ of CRE-positive cells were mCherry-positive, which in our opinion amounts to “substantial colocalization”.

In Fig. 2d, what are the units of Y axes? Should they be min/h and bouts/h? Are those hourly averages of the entire dark phase? If the units on Y axes are correct, WT spent in wakefulness for about 5 min x 8 times = 40 min on average in the entire dark phase, which sounds odd.

Thank you for pointing at this mistake. The bout number is /h which is now added to the Y axis.

In Fig. 2h, 2i, and 3, why are the measures close between Hcrt KO/KO and DKO but different in 5HTT +/-KO; Hcrt KO/KO? Generally speaking, the phenotype of heterozygous mice should be in the range between those of WT and homozygous mice.

In general, we agree with this comment, however recently the use of heterozygous mice, especially for monoamine research, has been proposed. It has been shown that for each individual phenotype, heterozygous (compared with knockout) mutants can display unaltered, partial-intermediate, enhanced or principally new phenotypes (1). In the 2 following publications (references 2 and 3) heterozygous animals demonstrated exaggerated behavior as compared to WT and KO counterparts, which we now discuss and add to our discussion section, as follows.

- 1- Kalueff et al. The developing use of heterozygous mutant mouse models in brain monoamine transporter research.
- 2- Montañez et al, Exaggerated effect of fluvoxamine in heterozygote serotonin transporter knockout mice.
- 3- GOGOS et al, Catechol-O-methyltransferase-deficient mice exhibit sexually dimorphic changes in catecholamine levels and behavior.

The sentence

“Additionally, a large increase in theta power was found in REMS of $5HTT^{+/-KO}; Hcrt^{KO/KO}$ mice. The question remains as to how the removal of a single allele of 5HTT in HcrtKO/KO mice affects so markedly REMS theta power and causes the other state-specific spectral changes seen in this genotype? Whether site specific monoallelic expression of the Slc6a4 gene underlies these findings warrants further investigations. Our data at gene expression

level show that heightened 5HT tone reduces the normal expression of *Hcrt* gene, which is consistent with the inhibitory role of 5HT on HCRT neurons.”

Changed to

“Additionally, a large increase in theta power was found in REMS of *5HTT^{+/KO};Hcrt^{KO/KO}* mice. The question remains as to how the removal of a single allele of 5HTT in *Hcrt*KO/KO mice affects so markedly REMS theta power and causes the other state-specific spectral changes seen in this genotype? The exaggerated behavior of heterozygous animals as compared to WT and KO counterparts is documented (49-50) for some genes. Additionally, whether site specific monoallelic expression of the *Slc6a4* gene underlies these findings warrants further investigations. Our data at gene expression level show that heightened 5HT tone reduces the normal expression of *Hcrt* gene, which is consistent with the inhibitory role of 5HT on HCRT neurons.”

Besides, as discussed in the manuscript (line 210-214), the observation is paradoxical. In lines 265-268, the authors discussed that mechanisms underlying REMS latency, REMS duration, and REMS theta power are differentially regulated. But this just means that there is no clear and explainable correlation between genotypes and sleep phenotypes. It is very difficult to take an overall message from this study.

Our observation that there is no correlation in our 5 genotypes between latency, duration, and theta power of the REM sleep state does by nature indicate that these 3 traits are differentially regulated. This has informative value in itself, for example suggesting that transition to REM sleep (state induction) and state maintenance (duration) are under regulation of both orexinergic and serotonergic signaling but by mechanisms partially independent of each other. We do not elucidate these mechanisms here, we show that they are linked to 2 major neuromodulators.

Lines 345-347, In contrast to cataplexy suppression, a large increase in REMS was found in DKO mice, indicating a dissociation between REMS and cataplexy and suggesting that the two states are regulated by different mechanisms: Such differential regulation of REMS and cataplexy has been already suggested in Hasegawa et al., 2014.

The differential regulation of REMS and cataplexy has been suggested by several authors, among whom our report on the characterization of cataplexy in orexin KO mice, which indicated that the EEG characteristics of the transition to REMS (pre-REMS) and the entry into cataplexy are singularly different (Vassalli et al. Brain, 2013). We still believe that our data represent so far the most convincing evidence of such differential regulation (beyond simple suggestion). Nevertheless, we now clearly make reference to Hasegawa et al., 2014 at lines 345-347 (...as also suggested by Hasegawa et al., 2014).

Reviewers' Comments:

Reviewer #1:

Remarks to the Author:

The authors have substantially addressed my comments.